# Prefrontal attentional saccades explore space rhythmically

Corentin Gaillard [1], Sameh Ben Hadj Hassen[1], Fabio Di Bello [1], Yann Bihan-Poudec [1], Rufin VanRullen [2,3] & Suliann Ben Hamed [1✉]

Recent studies suggest that attention samples space rhythmically through oscillatory interactions in the frontoparietal network. How these attentional fluctuations coincide with spatial exploration/displacement and exploitation/selection by a dynamic attentional spotlight under top-down control is unclear. Here, we show a direct contribution of prefrontal attention selection mechanisms to a continuous space exploration. Specifically, we provide a direct high spatio-temporal resolution prefrontal population decoding of the covert attentional spotlight. We show that it continuously explores space at a 7–12 Hz rhythm. Sensory encoding and behavioral reports are increased at a specific optimal phase w/ to this rhythm. We propose that this prefrontal neuronal rhythm reflects an alpha-clocked sampling of the visual environment in the absence of eye movements. These attentional explorations are highly flexible, how they spatially unfold depending both on within-trial and across-task contingencies. These results are discussed in the context of exploration-exploitation strategies and prefrontal top-down attentional control.

[1] Institut des Sciences Cognitives Marc Jeannerod, CNRS UMR 5229, Université Claude Bernard Lyon I, 69675 Bron, France. [2] Centre de Recherche Cerveau et Cognition, Université Paul Sabatier, UMR 5549, 31062 Toulouse, France. [3] Artificial and Natural Intelligence Toulouse Institute, Université de Toulouse, 31062 Toulouse, France. ✉email: benhamed@isc.cnrs.fr

The brain has limited processing capacities and cannot efficiently process the continuous flow of incoming sensory information. Selective attention allows the brain to overcome this limitation by filtering sensory information on the basis of its intrinsic salience (a child crossing the road in front of your car) or its extrinsic value (your old favorite coffee mug which you know is somewhere on your crowded desk). Visual selective attention speeds up reaction times[1,2], enhances perceptual sensitivity and spatial resolution[3,4] and distorts spatial representation up to several degrees away from the attended location[5]. Visual selective attention modulates both neuronal baselines[6,7] and the strength of visual responses[8], decreases neuronal response latencies[9], modifies the spatial selectivity profiles of the neurons[10,11] and decreases shared inter-neuronal noise variability[12].

Based on the early work of William James (1890)[13], the spotlight theory of attention assumes that attention is focused at one location of space at a time[1,14]. In this framework, the spotlight is moderately flexible. It is shifted from one location to another, independently from eye position, under the voluntary control of the subject, and its size is adjusted to the region of interest very much like a zoom lens. Converging evidence demonstrate that the prefrontal cortex (PFC) is at the origin of the attentional control signals underlying the behavioral attentional spotlight[7,15–18]. Supporting this idea, we recently demonstrated that this attentional spotlight can be reconstructed and tracked from PFC neuronal population activity with a very high spatial and temporal resolution[19,20]. However, recent experimental work provides a completely different perspective onto selective attention, suggesting that spatial attention samples the visual scene rhythmically[21–27]. These studies report that target-detection performance at an attended location fluctuates rhythmically very much like overt sampling processes, such as eye exploration in primates[28–30] or whisking in rodents[31]. The neural processes at the origin of this rhythmic sampling of space by attention are still poorly understood. Recent works propose that neural oscillations in the fronto-parietal network organize alternating attentional states or shifts in attention that in turn modulate perceptual sensitivity[23,32,33].

In the present study, we provide evidence reconciling these two seemingly contradictory views of spatial attention. Specifically, we demonstrate that the decoded PFC (x,y) attentional spotlight explores space continuously, through a sequence of attentional shifts that are generated at a specific alpha 7–12 Hz frequency. Crucially, we show that these oscillations of the attentional locus determine both neuronal sensory processing, defining how much

information is available in the PFC about incoming sensory stimuli, and perception, determining whether these incoming sensory stimuli are prone to elicit an overt behavioral response or not. Using Markov chain probabilistic modeling, we further show that space exploration by alpha-clocked attentional shifts depends on both trial and task specific spatial contingencies, implementing an alternation between exploration and exploitation cycles.

## Results

In order to access FEF attentional content in time, monkeys performed a manual response cued target-detection task (Fig. 1a) while we recorded the MUA bilaterally from their FEF neuronal ensembles, using two 24-contacts recording probes (Fig. 1c). Distractors (Fig. 1b, Supplementary Fig. 1) were presented during the cue-to-target interval and target luminosity was adjusted so as to make the task difficult to perform without orienting attention. Supplementary Fig. 2 and Supplementary Note 1 report MUA spatial attention selectivity. Previous studies demonstrate that PFC based decoding procedures allow to access in which quadrant[34–36] or at which (x,y) location attention is placed[19]. In these studies, neuronal signals were averaged over time intervals ranging from 150 to 400 ms (refs. [19,36–38]). Larger averaging window sizes produce higher decoding accuracies (Supplementary Fig. 3) but also result in the smoothing of dynamic changes in the spatial position of attention, artificially reinforcing a static view of the attentional spotlight.

**Prefrontal attention information oscillates at alpha rhythm.** Here, we seek to characterize spatial attention dynamics in time. The continuous decoding of attention is performed onto neuronal responses averaged over 50 ms successive time windows (1 ms time steps, Supplementary Fig. 3). At this temporal resolution, clear variations in the PFC attention-related information are observed. Indeed, when a classifier is trained to decode attention at a given time from cue onset, and tested onto novel activities recorded during the cue to target interval (cross-temporal decoding analysis, Fig. 2a), fluctuations in instantaneous classification accuracies can be noted, at a distance from cue processing. These fluctuations are reliably associated with a distinct peak in the power spectrum in the 7–12 Hz range (Supplementary Fig. 4 and Supplementary Note 2 for a discussion impact of averaging filter on decoded signal frequency content). Figure 2b shows an exemplar session. The power spectrum was quantified onto independent session time series (700–1200 ms following cue

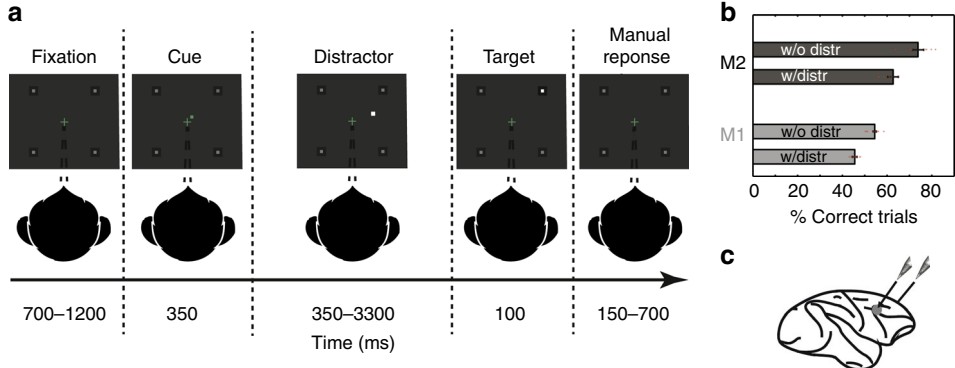

**Fig. 1 Task design and associated behavioral performance. a** 100% validity cued target-detection task with distractors. To initiate a trial, monkeys had to hold a bar and fixate a central cross on the screen. Monkeys received a liquid reward for releasing the bar 150–750 ms after target presentation. Target location was indicated by a cue (green square, second screen). Monkeys had to ignore any uncued event. **b** Behavioral performance of monkeys M1 and M2 at detecting the target in the presence (w/) or absence (w/o) of a distractor (median % correct +/− median absolute deviation, dots correspond to individual session data points). **c** Recording sites. On each session, 24-contact recording probes were placed in each FEF.

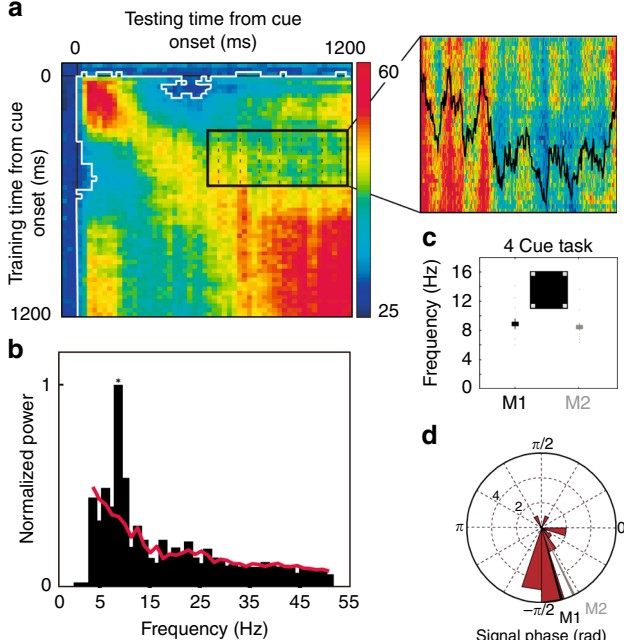

onset, Fig. 2a, inset), and assessed against the 95% confidence interval (random permutation, see methods, Fig. 2b, red line). On this session, peak frequency is identified at 9.2 Hz. Overall, PFC attention-related information oscillated, in monkey M1 (resp. M2), at an average frequency of 9 Hz (Fig. 2c, resp. 8.6 Hz, see Supplementary Fig. 5 for average normalized power spectrum across session and discussion of main alpha and lower theta peak). A clear phase-locking between these attention-related oscillations and cue onset can be seen across both monkeys (Fig. 2d, M1: −75°; M2 −65°). This rhythmic oscillation of the PFC attentional spotlight is thus phase reset by and actually pre-exists to cue presentation (see below). Importantly, oscillations can be identified from unilateral cortical recordings, in the same frequency range (Supplementary Fig. 6c). These oscillations are however in anti-phase between left and right hemispheres (Supplementary Fig. 6d), suggesting an active inter-hemispheric coordination mechanism (Supplementary Note 3).

**Alpha rhythm paces FEF population code.** Oscillations in the attention-related population activity can either reflect a global rhythmic entrainment of the entire FEF population (i.e., all neurons throughout the FEF, changing their firing rates coherently) or changes in the FEF population code at a specific frequency (i.e., only some FEF neurons changing their firing rates, at any peak or trough of the identified oscillations, each specific neuronal combination corresponding to a specific spatial attentional code). Figure 3a represents, for one recording probe, on an exemplar trial, and for each recording channel, the time epochs at which spiking-rate exceeds the 65% of the maximum spiking regime of the individual channel. On every single channel, these high spiking probability epochs do not appear to follow a systematic rhythm, thus contradicting a global rhythmic entrainment hypothesis. Rather, this high spiking probability organizes in discrete epochs, distributed over all recording channels. The hypothesis that changes in the FEF population code (and thus high spiking probability epochs) take place at a specific frequency implies that average MUA over all channels on a given trial will show marked rhythmic variations in firing rate in time. Figure 3b

**Fig. 2 Oscillation of prefrontal attention-related information. a** Cross-temporal classification around cue presentation ([−100: 1200 ms], step of 10 ms, averaging window of 50 ms) for an exemplar session. White contour: 95% confidence interval as assessed from trial identity random permutation. Black contour: close-up of the cross-temporal classification ([testing time: 500–1150 ms] post-cue; training time: [400–575 ms]) and corresponding mean classification along testing time (black). **b** Normalized power in this cross-temporal classification interval, (red line: 95% confidence interval) for the exemplar session presented in (**a**). **c** Average +/− s.d. of peak power in a 7–12 Hz interval, over all sessions, for each monkey (M1: black, M2: gray), in the 4-cued locations task version, individual data points plotted in red. **d** Circular distribution of signal phase with respect to cue onset (red bars), at identified peak frequency (mean phase: M1: black, −75°; M2: gray, −65°).

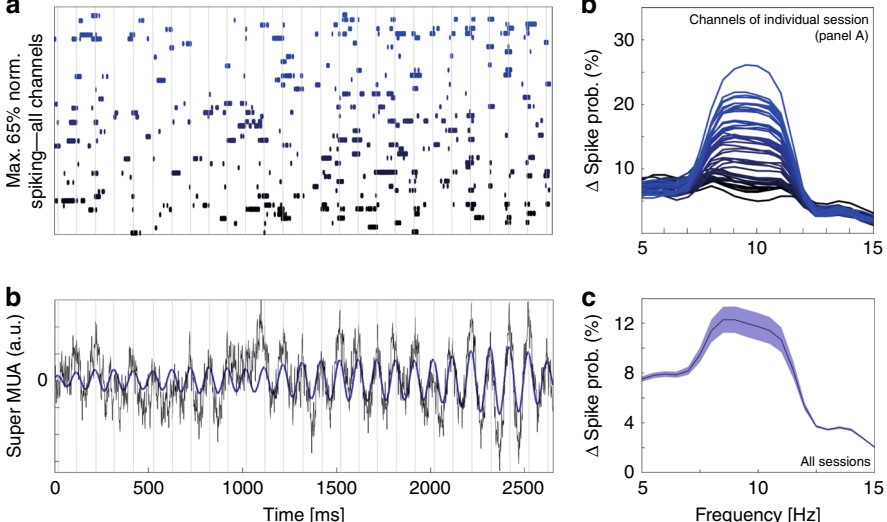

**Fig. 3 Alpha rhythm paces FEF population code. a** Individual channel spiking probability at a threshold of 65% (1 trial, 48 channels) in time. (Cue is presented at 700 ms. Gray vertical lines: peak of alpha cycles of the super MUA in (**b**). Individual channels ordered and color coded in a gradient of blue, as a function alpha locking amplitude in (**c**). **b** Raw (black trace) and alpha filtered single trial population super MUA calculated over the 48 MUA channels (blue trace). Gray vertical lines: peak of alpha cycles of the super MUA. **c** Changes in individual channel spiking probability, across all trials, as a function of putative locking to frequencies from 5 to 15 Hz. Spiking probability is specifically affected in the alpha frequency. Channels color coded in a gradient of blue, as a function of alpha locking amplitude. **d** Mean +/− s.e. phase frequency modulation of spiking activity across all sessions and all channels.

confirms this hypothesis. For this individual trial, a *super MUA* signal was computed by averaging the spiking activity of the 48 recording channels on this specific trial. Peaks of the alpha oscillations are clearly identified on the super MUA of the same individual trial[39] (Fig. 3b,) and plotted against the spiking probability changes represented in Fig. 3a. The high spiking probability epochs of individual channels coincide with peak alpha oscillatory phases in the super MUA. This is captured by a spectral analysis of changes in spiking probability in a frequency range running from 5 to 15 Hz. Most channels of Fig. 3a display a modulation of spiking probability in an 8–12 Hz frequency range (Fig. 3c, color code matching Fig. 3a). This holds true for all sessions (Fig. 3d, mean +/– s.e.). However, this alpha modulation of spiking probability does not reflect a global entrainment of the entire population. This can be seen in Fig. 3a in which individual channels do not exhibit on any given trial, high spiking rate synchronously at each identified alpha cycle. This is also captured in Fig. 3c, as the degree of alpha locking of spiking activity varies from one channel to the next. Rather, the channels with highest change in normalized spiking activity change from one super MUA alpha peak to the next, thus reflecting a change in the FEF population code. These variations correspond to changes in the spatial allocation of the attentional spotlight that will be described hereafter. Super MUAs independently computed across left and right electrodes on individual trials oscillated at a common rhythm (Supplementary Fig. 6a) as well as at a common phase (Supplementary Fig. 6b, Supplementary Note 3). In contrast, decoded population attention information from left and right probes oscillated at a common rhythm (Supplementary Fig. 6c), but in anti-phase one with respect to the other (Supplementary Fig. 6d). This confirms that these variations in MUA spiking probability correspond to changes in the spatial allocation of the attentional spotlight and suggest an active inter-hemispheric coupling mechanism. Importantly, the identified alpha clocking

frequency range didn't vary between superficial and deep cortical layers, indicating a common mechanism (Supplementary Fig. 7 and supplementary Note 4). However, alpha clocking power was higher in deeper layers as compared to superficial layers, possibly suggesting an origin in deeper cortical layers.

Previous studies indicate a coupling between LFPs theta and behavior[23] as well as between LFP beta and spiking activity and behavior[33]. An important question is thus whether changes in super MUAs are linked to the phase of oscillatory activity in the local field potentials. A significant alpha component is present in the super MUAs (Supplementary Fig. 8b) but not in the LFPs (Supplementary Fig. 8a). However, a significant phase-phase coupling can be identified between these two signals, in the alpha range (7–12 Hz) as well as in the beta frequency range (18–30 Hz, Supplementary Fig. 8d, Supplementary Note 5). The functional significance of this coupling, its directionality and its causal relationship to attention and perception remains to be explored.

**Attention oscillations predict target encoding and detection.** In order to quantify the link between PFC attention-related oscillations and both target processing and detection, trials were classified, for each session, as a function of when the target or the behavioral response were presented relative to the PFC attention information oscillation peak (Fig. 4a). In each session, oscillations were thus modeled by a sinusoidal wave with the session's specific oscillatory frequency and cue phase-shift. Targets were assigned to phase bins of width of $2\pi/10$, covering an entire oscillation cycle.

For hit trials, we quantified how much target-related information was available in the PFC neuronal population as follows (Fig. 4b, c). Neuronal activities were averaged between 50–100 ms post-target and used to quantify the accuracy of a four-class classifier at assigning target location to the actual quadrant it was

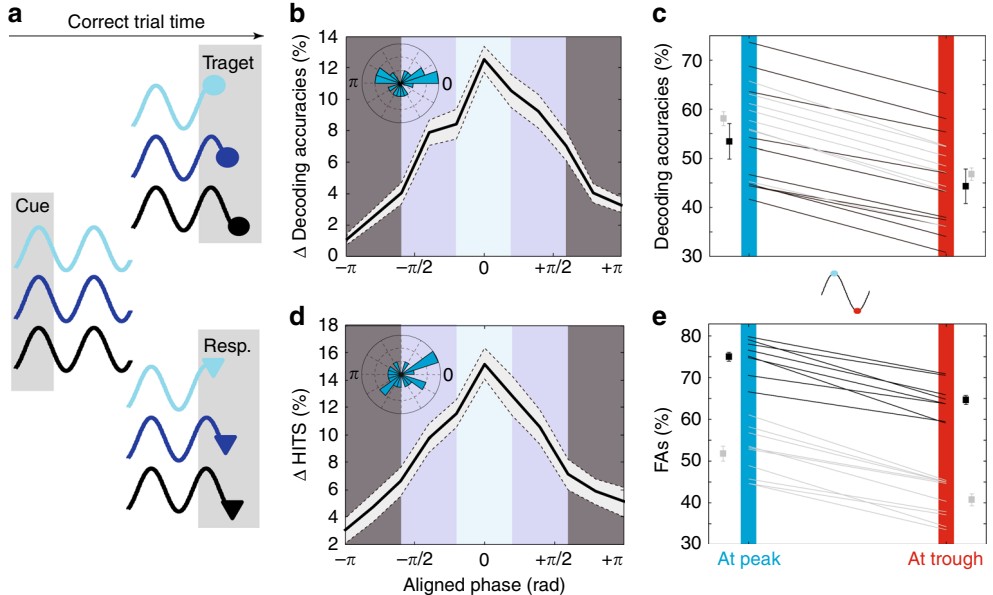

**Fig. 4 PFC target-related information and hit rates depend on when the TARGET is presented relative to attentional oscillation cycles. a** Categorization schema of trials as a function of when the target (top) and the behavioral response (bottom) were presented relative to attentional oscillation cycles. **b** Target related information as a function of phase with respect to the attentional oscillation cycles. Zero phase corresponds to optimal phase and not to zero phase-locking relative to the attentional signal, hence the observed phase shift distribution (inset: radial distribution of phase shifts relative to cue presentation, 18 sessions). **c** Peak (upper third of the distribution in (**b**)) to trough (lower third of the distribution in (**b**) variations (±s.e.) in target-related information, for each monkey (M1: black, M2: gray), for each session. **d** Percentage of hits is significantly higher in trials presented at optimal phase with respect to the attentional oscillation cycles (inset: radial distribution of phase shifts relative to cue presentation, 18 sessions). **e** Peak to trough variations in percentage of hits (±s.e.), for each monkey (M1: black, M2: gray), for each session.

presented in (see the Methods section). For each session, target-related PFC decoding accuracy was then computed for each independent bin of target-to-attentional oscillation phase-relationship. To increase the resolution of this analysis, this operation was repeated with successive phase bins shifted by 5% of their width. The lag that generated the highest discrimination between maximum and minimum decoding accuracy in the cycle was used to define optimal phase-shift between sensory processing and attention signal oscillations[40] (Fig. 4b). An average difference in peak and trough decoding accuracies of 10% can be noted when decoding accuracies are cumulated, across all sessions, at optimal phase-shift between sensory processing and signal oscillations (Fig. 4b). This difference is highly systematic as illustrated is Fig. 4c for each session and each monkey independently. The average target decoding accuracy at peak for monkey M1 (resp. monkey M2) was of 54% +/− 4 (resp. 58% +/− 2). At trough, these values dropped to 44% +/− 3 (resp. 47% +/− 1.5), in contrast with the low degree of inter-session variability that we report for PFC attention information locking to cue onset (Fig. 2d), phase lag between signal and optimal target processing was quite variable (Fig. 4b, inset). Overall, these results demonstrate a direct modulation of FEF target encoding by the ongoing alpha oscillations that we characterize on the PFC attention information.

We then used the same procedure as described above, in order to quantify, how target detection (hit rates) depended on target presentation time relative to the PFC attention information oscillation cycle (Fig. 4d, e). An average difference in peak and trough decoding accuracies of 10% can be noted when decoding accuracies are cumulated at optimal phase-shift between target detection and signal oscillations (Fig. 4d). Again, this difference is highly systematic as illustrated in Fig. 4e for each session and each monkey independently. The average target detection at peak for monkey M1 (resp. monkey M2) was of 75 +/− 1.5 (resp. 52% +/− 2). At trough, these values dropped to 64.5% +/− 1.5 (resp. 41.5% +/− 2). Two significant oscillatory peaks are observed onto hit rates relative to cue onset (Fig. 5a), one in the theta (3–5 Hz) frequency band, and one in the alpha (9–14 Hz) frequency band (Fig. 5b), thus reproducing previous behavioral observations[21–25,41]. These two peaks coincide with those identified in the prefrontal attention-related information (Supplementary Fig. 5), as well as with those identified in the FEF LFPs[23]. This alpha peak expresses in a frequency range that is lower that the FEF-theta locked alpha peak identified in the pulvinar[42]. Overall, as observed for target processing, we show a direct modulation of

behavioral target detection by the ongoing alpha oscillations in PFC attention information.

Although phase-lag between optimal target processing and optimal target detection (Fig. 4d, inset, Fig. 6a) was variable across sessions (Fig. 6b), mean reaction times, when cumulated over all sessions, significantly varied at a marked alpha rhythm (Figs. 6c, d; no theta is identified). In other words, alpha rhythm contributed both to an enhanced perception (hit rates) as well as to speeded up responses, both processes being probably coupled.

The above reported effects of PFC attention information oscillations onto target processing and behavioral outcome can either be interpreted in terms of modulations in attentional focus (i.e. attention dedication to sensory processing) or in terms of displacement of the attentional spotlight. In the following, we provide robust evidence in favor of attentional displacement.

**Attention rhythm predicts distractor encoding and detection.** Here, we explore the incidence of the oscillations in PFC attention-related information onto the processing of uncued distractors and the production of false alarms (Fig. 7), along the same experimental procedure used in the previous section to explore the incidence of the oscillations in PFC attention-related information onto the processing of cued targets and the production of hits. We first focused onto PFC distractor representation (Fig. 7b). An average difference in peak and trough distractor decoding accuracies from PFC neuronal responses of over 30% can be noted when decoding accuracies are cumulated at optimal phase-shift between distractor sensory processing and signal oscillations (Fig. 7b). This difference is highly systematic across sessions and monkeys (Fig. 7c). The average distractor decoding accuracy at peak for monkey M1 (resp. monkey M2) was of 45% +/− 2 (resp. 43% +/− 1.7). At trough, these values massively dropped to 14% +/− 3.5 (resp. 7% +/− 3). As observed for target processing, phase lag between signal and optimal distractor processing was quite variable (Fig. 7b, inset).

In a second step, we quantified how responses to distractors (false alarm) depended on distractor presentation time relative to the PFC attention information oscillation cycle. An average difference in peak and trough false alarm rate of more than 10% can be noted when false alarms are computed at optimal phase-shift between distractor detection and signal oscillations (Fig. 7d). This difference is highly systematic across sessions and monkeys (Fig. 7e). The average distractor detection at peak for monkey M1 (resp. monkey M2) was of 45% +/− 1.5 (resp. 42% +/− 2). At trough, these values dropped to 36% +/− 2 (resp. 29% +/− 2.5).

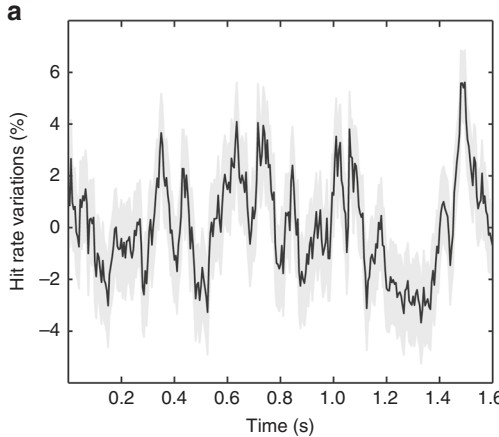
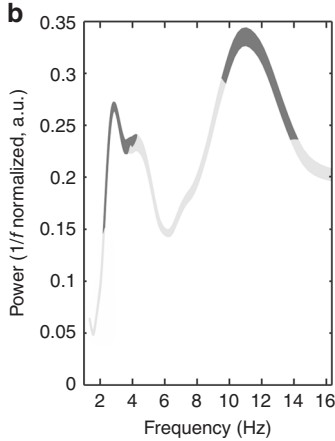

**Fig. 5 Oscillations in cumulated behavioral performance. a** Changes in hit rates (detrended mean +/− s.e.) as a function of time of target presentation relative to cue presentation. Behavioral data compiled across all recording sessions and both monkeys. **b** Behavioral time series power spectrum (mean +/− s.e.). Frequencies significantly modulating overall behavioral performance in dark gray (95% CI on random permutation of target timings).

As seen for hit rates, phase lag between signal and optimal distractor detection was quite variable (Fig. 7d, inset).

Overall, we show a direct modulation of how the PFC represents distractors as well as the overt behavioral responses to distractors by the ongoing PFC attention information alpha oscillations. These observations support the hypothesis of a displacement of attention in space. In the following, we provide evidence for an explicit link between the above described

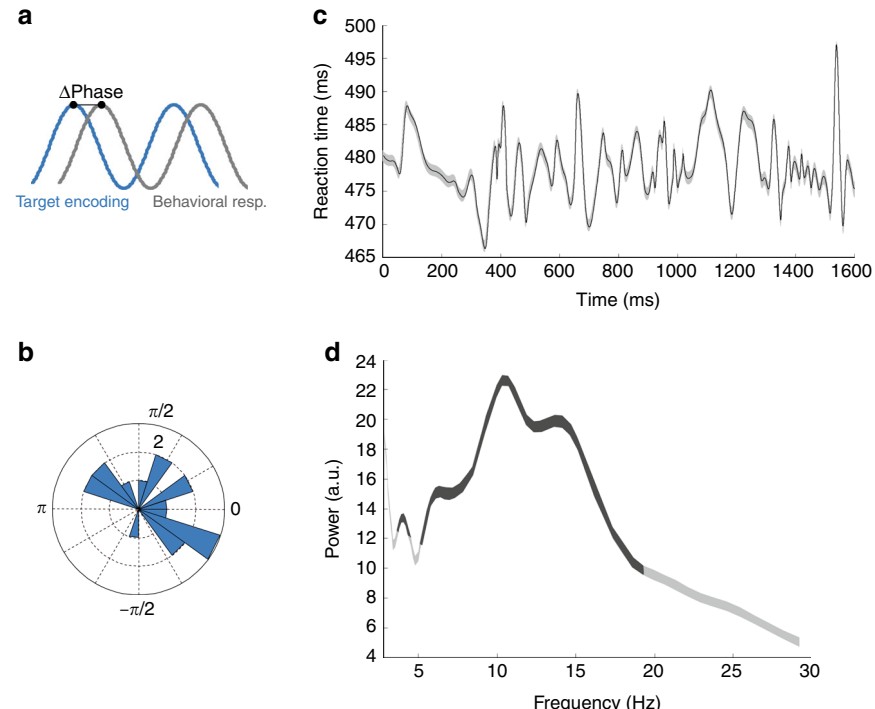

**Fig. 6 Phase lag between optimal target encoding and optimal target detection behavioral response. a** presents a session to session variability (**b**). **c** Changes in reaction times to target presentation as a function of time from cue presentation (detrended geometric mean +/− s.e.). **d** Reaction time series power spectrum (mean +/− s.e., dark gray shaded areas, significance w/ 95% CI).

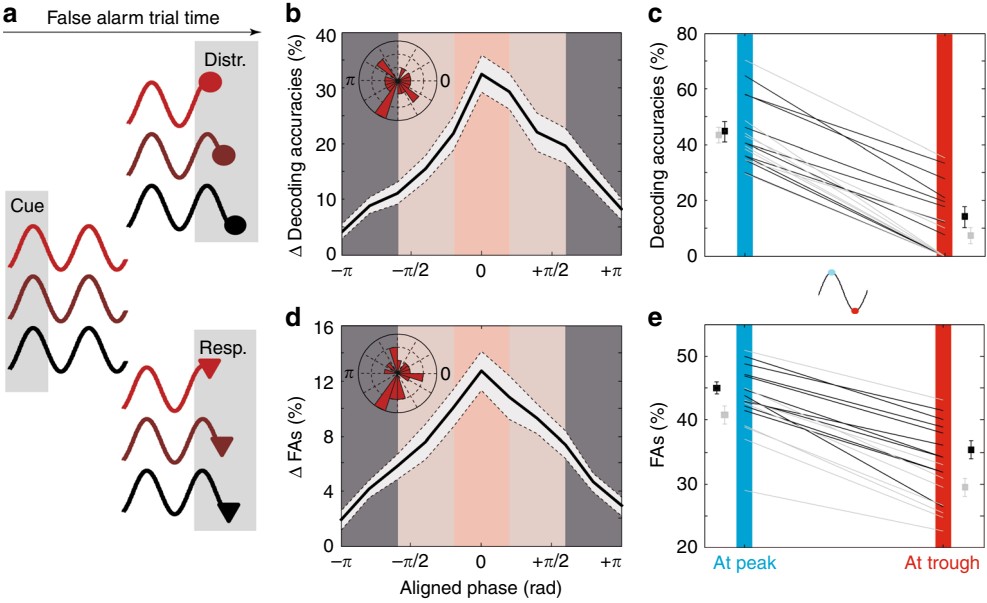

**Fig. 7 PFC distractor related information and false alarm rates depend on when the DISTRACTOR is presented relative to attentional oscillation cycles. a** Categorization schema of trials as a function of when the distractor (top) and behavioral response (bottom) was presented relative to attentional oscillation cycles. **b** Distractor related information is significantly higher in trials presented at optimal phase with respect to the attentional oscillation cycles. **c** Peak to trough variations in distractor related information (M1: black, M2: gray), for each session (±s.e.). **d** Percentage of false alarms is significantly higher in trials presented at optimal phase with respect to the attentional oscillation cycles (inset: radial distribution of phase shifts relative to cue presentation, 18 sessions). **e** Peak to trough variations in percentage of false alarms (M1: black, M2: gray), for each session (±s.e.). All else as in Fig. 3.

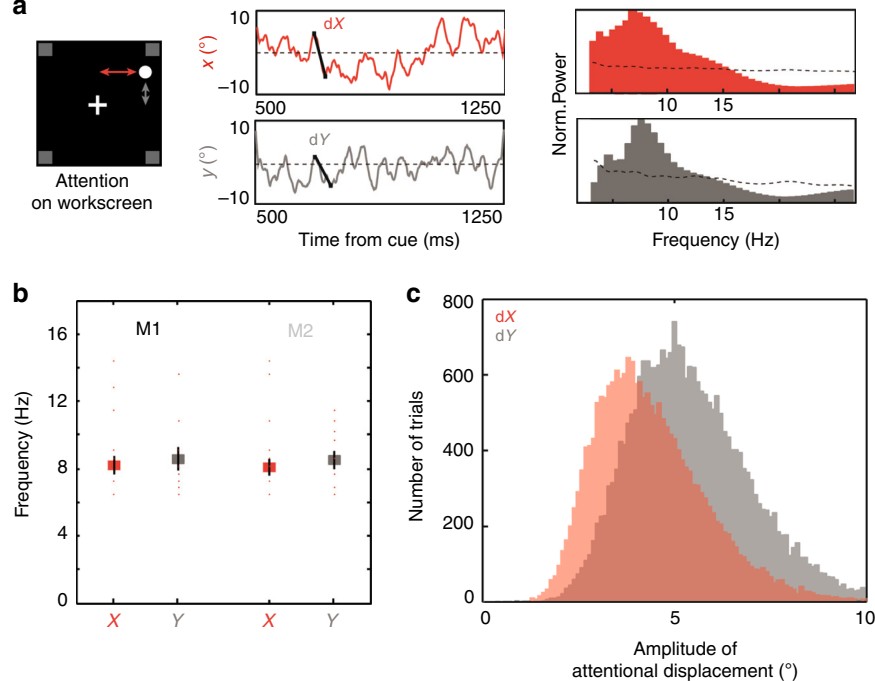

**Fig. 8 The spatial oscillations in PFC attentional information map onto changes in attentional spotlight position in space.** The (x,y) position of the attentional spotlight (**a**, left panel: x, red; y, gray) varies in time (**a**, middle panel) at a frequency (**a**, right panel, dotted black line, 95% confidence interval) that matches the frequency characterized in overall PFC attentional information (cf. Fig. 2). **b** Mean maximal power peak frequency of attentional spotlight trajectory for M1 and M2, in x (red) and y (gray), individual data points plotted in red. **c** Distribution of amplitude of attentional displacement during the cue to target interval (500–1250 post cue), along the x (gray) and in y (red) dimensions.

oscillations in PFC attention information and exploration of space by a dynamic and rhythmic attentional spotlight operating in the alpha frequency range.

**Attentional saccade-like exploration.** In a previous study[19], we demonstrated that the continuous (x,y) readout of a linear classifier assigning neuronal activities to a spatial location of attention in the PFC is predictive of behavior, both in terms of hit and false alarm rates. In the following, we apply the same approach to extract (x,y) attentional spotlight trajectories in time before and after cue presentation, to the major difference that the readout is obtained at higher temporal resolution, from neuronal responses averaged over 50 ms rather than on 150 ms as presented in the Astrand et al.[19]. Supplementary Movie 1 presents such PFC attentional spotlight trajectories for an exemplar trials. The attentional spotlight is not stable, nor is it hopping between the four most salient locations. Rather, it is exploring space through a succession of attentional displacement bringing the spotlight both around and away from the cue.

Projections of an exemplar PFC attentional spotlight trajectory onto the x- and y-dimensions are presented in Fig. 8a (middle panel), as well as their power spectrum (right panel). A systematic rhythm in attentional displacement can be identified on both x- and y-traces, on all trials and all sessions, for each monkey (Fig. 8b, monkey M1: $X = 8.1$ Hz +/ −1.6, $Y = 8.3$ Hz +/−2; monkey M2: $X = 8$ Hz +/ −1, $Y = 8.4$ Hz +/−2), in the same range as identified for the global attention population information. No statistical difference is observed between alpha oscillatory peaks in the x- and y- attentional traces and in the global attention information content ($p = 0.49$ and $p = 0.87$ respectively), confirming a strong link between these measures. These PFC attentional spotlight trajectories are exploring space homogenously. Interestingly, a significant difference was observed between the distributions of attentional displacement along the x-

and y-axis ($p < 0.0001$, Kolmogorov–Smirnov test), indicating a larger exploration of space along the vertical dimension. Overall, the PFC attentional spotlight explores space both rhythmically and continuously.

**Task variables define rhythmic attention deployment in space.** During cued target detection tasks, the cue orient attention towards the spatial location where the target is expected to be presented. The absolute distance between two successive attentional shifts does not vary between the period before (Fig. 9a, black) and after cue presentation (Fig. 9a, gray, Kolmogorov–Smirnov test $p > 0.99$). However, the spatial distribution of these shifts vary significantly between pre-cue and post-cue epochs. Specifically, Fig. 9b represents the heat maps of the spatial distribution of the decoded attentional spotlight during the pre-cue interval (−500 to –200 ms, contour 1) and the post-cue interval (500–1200 ms, contour 2), for each category of cued trials (T1, T2, T3, and T4). During the pre-cue epoch, the heat maps are centered onto the fixation point (median 0.9° +/− 0.07°), exploration being confined within the 10.7° central degrees. During the post-cue epoch, the heat maps shift towards the cued landmark by, on average, 3.6° (+/−0.2). For all cued conditions, attentional exploration, extends up to 14.5° towards the cued location (exploration probability threshold of 60%). We thus show that attentional exploration trajectories depend on trial structure.

The next question is thus whether the attentional temporal dynamics and attentional exploration trajectories described up to now are also influenced by task structure. To address this question, we use Markov chain probabilistic modeling to describe how the attentional spotlight explores space in two different versions of a cued target detection task, that only differed in the number and localization of the task-relevant items: a first version (above analysis), in which the cue could orient attention to one of

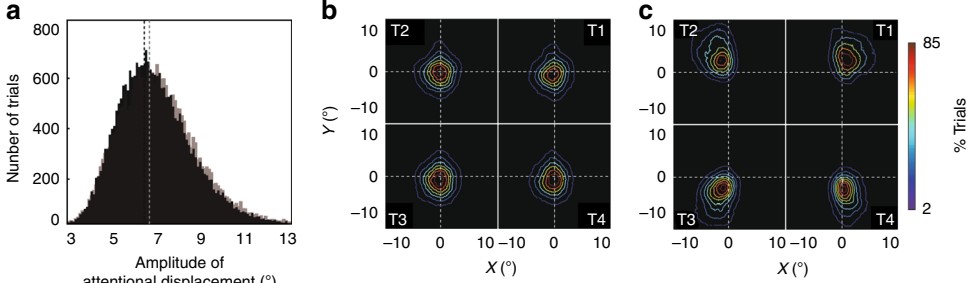

**Fig. 9 Attentional spotlight exploration is an endogenous process affected by task events. a** Distribution of amplitude of attentional displacement between one PFC attentional position and the next, in the pre-cue period (black) and in the cue-to-target interval (gray). PreCue (**b**) and postCue (**c**) heat maps of the spatial distribution of the decoded attentional spotlight.

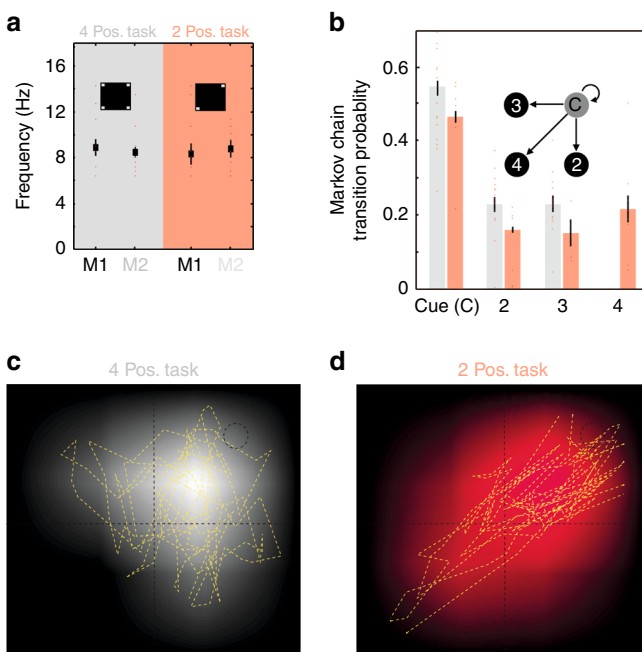

**Fig. 10 How PFC attentional spotlight rhythmically explores space depends on task context. a** 7–12 Hz oscillation peak in PFC attention information in a four (gray) or two (red) cued positions target detection task, for monkeys M1 (black) and M2 (gray), over all sessions (±s.e.), individual data points plotted in red. **b** Markov chain probability (±s.e.) of the attentional spotlight to stay at the cued quadrant [C], to transition to the same hemifield uncued quadrant [2], to transition to the opposite hemifield uncued quadrant [3] or to transition to the diagonally opposite uncued quadrant [4], in the four (gray) or two (red) cued positions target detection task. **c** Single trial example of PFC attentional spotlight exploration (10 ms resolution) during the cue to target interval, in a four cued positions target detection task, superposed onto the Markov chain probability map for this cued condition, individual data points plotted in red. Cue was presented in the upper right quadrant. **d** Same as in (**c**) for a two cued positions target detection task.

the four possible quadrants (18 sessions), and a second version in which the cue oriented attention to only two possible quadrants, placed along the diagonal one with respect to the other (16 sessions). For both monkeys, oscillations in the PFC attention information did not depend on the task (Fig. 10a). In contrast, how the decoded attentional spotlight was deployed onto space was drastically different between the two tasks. This is captured by the Markov chain probabilistic modeling of the probability of the spotlight to stay in the cued quadrant when already there, or

to shift to one of the uncued quadrants (Fig. 10b, see methods). Indeed, while during the two types of task configurations, the probability that the decoded attentional spotlight remained at the cued location was highest (probabilities of 0.55 and 0.47 respectively), probability pattern of attention transitioning from the cued location to one of the uncued quadrants was very distinct. Specifically, during the four position task, virtually no transitions between the cued quadrant and the diagonally opposite quadrant can be observed (Fig. 10b, gray, probability of transition of 0; for comparison, probability of transition from cued location to position 2: 22%; to position 3: 23%). This is exemplified in Fig. 10c, which represents the decoded attentional spotlight trajectory during the cue to target interval in a representative trial of a four position task. In contrast, during, the two position task, transitions between the cued quadrant and the diagonally opposite quadrant, the second most relevant spatial location in the task, become dominant with respect to the other two uncued quadrants (Fig. 10b, red, probability of transition of 22%; for comparison, probability of transition from cued location to position 2: 16%; to position 3: 15%). This is exemplified in Fig. 10d, which represents the decoded attentional trajectory during the cue to target interval in a representative trial of a two position task.

Overall, we provide evidence that the PFC attentional spotlight explores space at an alpha that remains stable within trials and across tasks. However, we show that how this decoded spotlight explores space depends on both within-trial and across-task task contingencies.

## Discussion

**The prefrontal attentional spotlight explores space rhythmically.** Converging behavioral evidence indicates that attention and perception are not anchored at a specific location in space, but rather exhibit a temporal alpha rhythmicity[26]. This rhythmic sampling of space is phase-reset and entrained by external events of interest. It can also be observed spontaneously[43], and is proposed to organize the tracking of task-relevant spatial locations by attention in time[21,22,24,26,27,41,44,45]. It has been proposed that, when prior information is available, such a rhythmic sampling of information is more efficient than a continuous sampling of space[46]. These observations have led to reconsider the model of a continuously active attention spotlight in favor of a rhythmic sampling of attention at relevant spatial locations, including during sustained attention states[22,26].

Our findings reconcile these two models, describing a dynamic attentional spotlight that continuously explores space at a specific rhythm. This rhythmic exploration shares major characteristics with previous behavioral reports: (1) these oscillations are ongoing and can be identified independently of the intervening task events, (2) they are reset by relevant external events such as

spatial cues and (3) they occur in a well-defined functional alpha frequency range. However, even if attentional exploration targets task-relevant locations, as reflected by the rhythmic enhancement of neuronal sensory processing and behavioral performance at the cued target location, exploration is not restricted to these locations. Rather, space exploration by attention extends to uncued task irrelevant spatial locations, as reflected by the rhythmic enhancement of neuronal sensory processing and behavioral overt report at uncued unpredictable distractor locations. Several recent behavioral studies suggest that attention fluctuates at around 8 Hz. This sampling can be distributed across multiple spatial locations[24,25] or multiple objects at a given location[47]. Here, we argue that the decoded attentional spotlight provides a direct access to the intrinsic attentional rhythm, i.e., 8 Hz, though how this reflects onto behavior will fully depend on task design and on how the spotlight successively samples relevant task locations. In our task, cues have a 100% validity. Hence, the attentional spotlight is on average into the cued quadrant, sampling visual information at 8 Hz. We predict that if the cue was not fully valid, behavioral sampling frequency might be lower than 8 Hz, directly co-varying with cue validity. Attentional periodicity in monkey FEF has been suggested before, albeit with a faster frequency[33] (18–25 Hz). This frequency difference may reflect disparities in experimental design or task difficulty. Alternatively, it could reflect specific differences between FEF LFP and MUA processes (Supplementary Fig. 8a). The present study goes beyond this prior work by explicitly decoding the position of attention over time, and exploring the effect of distractors and task contingencies.

The phase between the attentional spotlight ongoing oscillations and a given stimulus presentation accounts from 10% (in the case of the target) to 30% (in the case of distractors) of the accuracy with which PFC neuronal populations encode the location of this stimulus. In other terms these oscillations—i.e., where the attention spotlight falls in space—critically impact the sensory processing of incoming stimuli. Tracing down this effect all throughout the visual system would be extremely relevant. Neuronal responses to low-salience task-relevant stimuli has been shown to arise earlier in the PFC than in the parietal cortex[7]. As a result, one predicts that this dependence of sensory processing onto attention spotlight oscillations will be found at all stages of the visual system. However, phase relationships between local neuronal and stimulus presentation is expected to vary, reflecting a top-down cascade of influences, in agreement with the role of the FEF in attentional control[7,16–18,48,49].

These oscillations also determine overt behavioral perceptual outcome, accounting from 10% (in the case of false alarm production) to 30% (in the case of correct target identifications) of stimulus detection. This is globally higher than the range of reported oscillatory changes in behavioral hit rates[22,23,25], highlighting the high predictive power of these neuronal population oscillations.

Overall, this suggests the existence of perceptual cycles[26] that organize as a rhythmic alternation between exploitation and exploration states of space sampling by attention[50].

**Exploring versus exploiting space by attention**. Two models have been proposed to account for the spatial deployment of attention[51–55] a parallel processing model, driven by bottom-up information, dominating when visual search is easy; and a serial processing model, driven by top-down mechanisms, dominating in difficult visual search[41]. In the context of this latter model, it has been hypothesized that the brain controls an attentional spotlight that scans space for relevant sensory information. In a previous study[19], we assessed, based on the (x,y) decoding of the neuronal population activity of the FEF, the tracking of this attentional spotlight in time[19]. Here, we show that the attentional spotlight explores space serially both at relevant (cued) and irrelevant (un-cued) locations, alternating between the exploitation and the exploration of the visual scene[26]. The activity of the parietal[10] and PFC[56] cortical regions has been shown to change drastically between exploitation and exploration behavior. In particular, exploration is associated with faster though less accurate oculomotor behavior[10] and a disruption of PFC control signals[56]. This is proposed to facilitate the processing of unexpected external events[10], the expression of novel behavior and learning through trial and error[56].

Fiebelkorn and Kastner[50] propose that theta rhythms organize neural activity into alternating attentional states associated with either sampling (coinciding with periods of enhanced perceptual sensitivity) at a behaviorally relevant location or shifting to another location (coinciding with periods of decreased perceptual sensitivity). In this model, how much overt or covert attention is placed onto a given item of the visual scene depends on its behavioral relevance. Because in our task, exploitation is an unexpected low frequency event, we propose that exploration is the default mode of the system, while exploitation, requires effort or a top-down drive to be implemented. Whether this exploitation is implemented by an independent theta clock remains to be tested. This would reconcile the seemingly contradictory views of the sampling/shifting hypothesis and an alpha exploration/exploitation hypothesis (see Supplementary Note 7 for a thorough discussion of this point).

Our observations strongly indicate that exploration and exploitation dynamically alternate within trials. This alternation of exploration and exploitation of space by the attentional spotlight thus appears to optimize subject's access to incoming information from the environment by a continuous exploration strategy, very much like is described for overt exploration behaviors such as saccadic eye movements, whisking or sniffing[25,57,58]. This covert exploration of the environment by attention however takes place at a slightly higher frequency than the typical theta exploration frequency described for overt exploration. This is probably due to energetic and inertial considerations in controlling the remote effector during overt exploration (e.g., eye, whisker or nose muscles). Interestingly, the rhythm at which thisexploration/exploitation alternation takes place coincides with the rhythm at which attention behaviorally explores the different part of a given object[22]. Overall, this leads us to postulate the existence of attentional saccades that can either be directed towards specific items for exploitation purposes, or deployed onto the entire visual field for exploration purposes.

**Continuous attentional sampling and attentional saccades**. Covert exploration of space by attention is more energy efficient than overt exploration by the eyes and the former serves to inform and guide the latter. In an initial "premotor theory of attention", these two processes, namely attentional selection and saccadic eye movements, have been suggested to rely on identical cortical mechanisms. This theory hypothesizes that attentional displacements, mirror saccades of the eyes except for the recruitment of the extra-ocular muscles[59]. Since then, several studies have demonstrated a functional dissociation between these two processes, and rhythmic attentional sampling has been shown to be independent from micro saccade generation[23,25,60]. Our observations support a continuous exploration of space by the attentional spotlight organized thanks to a rhythmic re-orientation of the attentional spotlight taking place at an alpha rhythm. This framework leads to an interesting set of experimental predictions. For instance, attentional capture and

distractibility by an intervening distracting item is expected to coincide with an ongoing attentional re-orienting towards this item[19]. Likewise, inhibition of return[61–64], is expected to reflect as an under-exploration of previously visited locations with respect to unexplored locations. This covert saccade-like exploration is proposed to be an intrinsic property of attention, taking place irrespectively of the ongoing behavior and building onto a rhythmic alpha clock. Its spatial pattern, that is to say the portion of space that is being explored by these attentional shifts, as well as the frequency at which task-relevant items are explored are however expected to be under top-down control.

**Top-down control**. Numerous studies indicate that PFC and specifically the FEF play a crucial role in attention orientation and attention control[7,16–18,35,48,49]. As a result, one expects that the exploration of space by the PFC attention spotlight be strongly biased by top-down voluntary control. Confirming this prediction, we show that task goals significantly affect attentional space exploration strategy. Specifically, the locations where the PFC attentional spotlight explores space are modulated both (1) within trials, by the expected position of the target after cue presentation, and (2) across tasks, by the general expectations about sensory events. In other words, relevant task items are more explored than irrelevant locations, where relevance concatenates information relative to the ongoing trial and task design. This is in agreement with prior behavioral observations reporting that the attentional sampling rate observed at the behavioral level decreases as the number of task-relevant items increases[24,65]. Overall, this indicates that the rhythmic exploration of space by attention, is an intrinsic, default-mode state of attention, that can be spatially modulated by task context and internal expectations. A strong prediction is that this rhythmicity in attentional spatial processing will directly impact attention selection processes in lower level cortical areas, through long-range feedback processes[60], possibly mediated by NMDA receptors[66].

Overall, our work describes for the first time the spatial and temporal properties of the population PFC attention spotlight. It demonstrates a continuous exploration of space, that is mediated by attentional saccades that unfold at an alpha 7–12 Hz rhythm and that accounts for both neuronal sensory processing reliability and overt behavioral variability. Importantly, it bridges the gap between behavioral evidence of attentional rhythmic space sampling and local field attention-related oscillatory mechanisms[23,26,32], revealing the neuronal population dynamics associated with rhythmic attentional sampling.

## Materials and methods

**Behavioral task and experimental setup**. The task is a 100% validity endogenous cued target detection task (Fig. 1a). The animals were placed in front of a PC monitor (1920 × 1200 pixels and a refresh rate of 60 HZ), at a distance of 57 cm, with their heads fixed. The stimuli presentation and behavioral responses were controlled using Presentation (Neurobehavioral systems®, https://www.neurobs.com/).
To start a trial, the bar placed in front of the animal's chair had to be held by the monkeys, thus interrupting an infrared beam. The onset of a central blue fixation cross (size 0.7° × 0.7°) instructed the monkeys to maintain eye position inside a 2° × 2° window, defined around the fixation cross. To avoid the abort of the ongoing trial, fixation had to be maintained throughout trial duration. Eye fixation was controlled thanks to a video eye tracker (Iscan™). Four gray square landmarks (LMs—size 0.5° × 0.5°) were displayed, all throughout the trial, at the four corners of a 20°x20° hypothetical square centered onto the fixation cross. Thus, the four LMs (up-right, up-left, down-left, down-right) were placed at the same distance from the center of the screen having an eccentricity of 14° (absolute x- and y-deviation from the center of the screen of 10°). After a variable delay from fixation onset, ranging between 700 and 1200 ms, a small green square (cue - size 0.2° × 0.2°) was presented, for 350 ms, close to the fixation cross (at 0.3°) in the direction of one of the LM. Monkeys were rewarded for detecting a subtle change in luminosity of this cued LM. The change in target luminosity occurred unpredictably between 350 and 3300 ms from the cue off time. In order to receive a reward (drop of juice), the monkeys were required to release the bar in a limited time window (150–750 ms) after the target

onset (Hit trial). In order to make sure that the monkeys did use the cue instruction, on half of the trials, distractors were presented during the cue to target interval. Two types of distractors could be presented: (i) uncued landmark distractor trials (33% of trials with distractor); these corresponded to a change in luminosity, identical to the awaited target luminosity change, and could take place equiprobably at any of the uncued LMs; (ii) workspace distractor trials (67% of trials with distractor); these corresponded to a small square presented randomly in the workspace defined by the four landmarks. The contrast of the square with respect to the background was the same as the contrast of the target against the LM; when presented at the same radial eccentricity as the LMs, the workspace distractor had the same size as the landmarks; for smaller eccentricities, the size of the workspace distractor was adjusted for cortical magnification such that it activated an equivalent cortical surface at all eccentricities. The monkeys had to ignore all of these distractors. Responding to any of them interrupted the trial. If the response occurred in the same response window as for correct detection trials (150–750 ms), the trial was counted as a false alarm (FA) trial. Failing to respond to the target (Miss) similarly aborted the ongoing trial. Overall, data was collected for 19 sessions (M1 10 Sessions, M2 9 Sessions). The behavioral performance of each animal is presented in Fig. 1b, for hit, miss and false alarm trials. In order to characterize whether the attentional temporal dynamics and attentional exploration trajectories described in this study were influenced by task structure, a second two-position variant of the above described task was also presented to the monkey. In this task, while the four landmarks were present all throughout the task as previously, only two diagonally opposite positions amongst the four were cued all throughout the session. The pair of cued stimuli changed from one session to the next. 16 such sessions were recorded (eight sessions for M1, eight sessions for M2). All else was as described for the main four position task.

**Electrophysiological recording**. Bilateral simultaneous recordings in the two frontal eye fields (FEF) were carried out using two 24 contacts Plexon U-probes (Fig. 1b). The contacts had an interspacing distance of 250 μm. Neural data was acquired with the Plexon Omniplex® neuronal data acquisition system. The data was amplified 400 times and digitized at 40,000 Hz. A threshold defining the multi-unit activity (MUA) was applied independently for each recording contact and each session before the actual task-related recordings started.

**Neuronal decoding procedure**. MUA recorded during the task were aligned on the cue presentation time and sorted according to the monkey's behavioral response (Correct trials, misses trials, false alarm trials). As in Astrand et al.[19,35], a regularized linear decoder was used to associate, on correct trials, the neuronal activity estimated on a given interval in the cue to target interval and the cued location. The decoder was trained on a random set of 70% of the correct trials at a specific time in the cue to target interval, then tested on the 30% remaining at all time after cue presentation (see Supplementary Note 6 for a discussion of how classical decoding techniques apply to the decoding of a dynamic attentional spotlight as described here). During training, the input to the classifier was a 48 elements by N-trial matrix corresponding to the average neuronal response on each recording channel for the time interval of interest for each of the N training trials. The imposed output of the classifier was the (x,y) coordinates of the cued landmark for each of these N training trials. During testing, the output of the classifier was estimated for a 48 element vector corresponding to the average neuronal response on each recording channel for the time interval of interest on a testing trial, new to the classifier. This output can be read as a continuous (x,y) estimate of attention location[19] or as a class output, corresponding to one of the four possible visual quadrants[19,34,35]. When seeking for a continuous (x,y) readout of attention location, we performed the training using the neuronal activities of Hits averaged over 50 ms immediately before target presentation, then we tested the decoder on neuronal activities averaged over 50 ms all throughout the cue to target interval. When taking a classification perspective, we performed cross-temporal decoding analyses (supplementary fig. 3ab), where successive classifiers were formed based on successive overlapping (every 10 ms) time windows during the cue to target interval and tested on independent trials and successive overlapping time windows during the cue to target interval. Mean decoding performance was calculated along the testing axis as the number of correct classifications divided by the total number of classifications. This procedure was repeated 10 times and the grand average over the 10 repeats are used for further analyses. Supplementary fig. 3c-h represents this cross-temporal decoding analysis performed onto a training and a testing time interval running from cue presentation to 1200 ms post-cue, when the classifiers are based on neuronal activity sampled over 300, 150, 100, 75, 50 or 25 ms. As expected, overall classification performance drops with neuronal sampling window size[67]. Importantly to the present paper, temporal variations in available content arise at lower sampling window sizes (Fig. 2, Supplementary Fig. 3f–h). The core analyses of the present paper were performed using a neuronal sampling window size of 50 ms.

**Oscillations in behavioral performance**. Hits and Misses from M1 and M2 were compiled in time (aligned to cue presentation), and merged together across the 19 recording sessions. Behavioral performance, defined as the proportion of (hits/(hits + misses)) was then computed at every millisecond over. The spectral analysis

of this time series was performed on detrended data using a Morlet Wavelet transform as in Fiebelkorn et al.[23], over the attentional period ranging from 500 ms post cue presentation to 2100 ms. Standard error in the power spectrum corresponds to spectral variability during this time interval. Global power spectrum 1/f component was removed from the dataset using a *f normalization (Fig. 5).

**Signal frequency and phase analyses**. In the present paper, frequency and phase analyses were performed onto time series (inset in Figs. 2a, b) representing attention information classification performance during cue to target interval, for a given training time, along a testing time running from 500 to 1200 ms from cue onset. Time series were evaluated at training times ranging from 500 to 1200 ms from cue onset, each time series representing a data sample. Frequency and phase analyses were performed using Wavelet Transform Analyses, based on the Wavelet Coherence Matlab Toolbox[68]. Specifically, for the time frequency analyses, Morlet wavelet transforms were independently applied to the original data time series (12 Octaves per scale). The significance of peak frequency distributions in the range of interest (7–12 Hz) was assessed against the frequency content of time series generated by the random permutation (1000 repetitions, Fig. 2b, dashed line) of the MUA time series (prior to decoding). Power to frequency plots are represented with a low frequency cutoff at 4 Hz and normalized by maximal spectrum value. Phase of the signal with respect to cue presentation were obtained from the complex wavelet transform of the signal at the peak frequency of each session.

**Impact of population oscillations onto individual MUAs**. For each trial, channel and session, spike trains were smoothed on a 50 ms sliding window over a −700 ms pre-cue to 2000 ms post-cue time series. On the one hand, a Super MUA signal was computed by averaging the spiking activity of the 48 recording channels of each session and each trial. On the other hand, the initial individual channel continuous spiking activity was transformed to identify high-spiking (defined by a spiking rate above 65% of the maximum spiking regime of the individual channel, labeled as 1) and low-spiking (labeled as 0) epochs. The probability of individual channel firing as a function of the oscillatory cycles of the session's Super MUA was then computed as follows. For each channel, for frequencies from 5 Hz to 15 Hz, the spiking probability was computed for the up ($+/-\pi/2$ around oscillation peak) and down ($+/-\pi/2$ around oscillation trough) oscillatory phases of the frequencies of interest over the entire time window. For each frequency, the analysis time window was adjusted to 1.5 oscillatory cycle length and computations were performed over a minimum of 50 time bins. All further analyses on this metric were performed onto an attentional epoch running from 500 ms post-cue to 2100 ms post-cue.

**Peak and trough classification**. In order to track whether the frequencies identified on the decoded attentional information causally reflected onto behavior, the following analysis was performed. For each session i, characteristic attention information oscillatory frequency F(i) and Phase P(i) determined using the above described wavelet transform analysis. The decoded classification attention information signal was modeled as a sinusoidal wave determined by the function $MSi(t) = \sin(2.\pi.F(i).t-P(i))$. Using this modeled signal (MSi), and based on target time from cue presentation, trials were assigned to one of 10 possible phase intervals ranging from $[-\pi + \pi]$ phase offset from the modeled sinusoidal wave For each of these subsets of trials, decoding accuracy of target location (resp. distractor location) and percentage of hit trials (resp. FA trials) was extracted (Fig. 3b, c and 4b, c). As sensory processing or behavioral outcome could be phase lagged with respect to signal oscillations, target time was progressively shifted using 5 ms steps, so that the phase interval associated with peak sensory processing or behavioral outcome coincided with phase 0. This procedure was applied independently for each of the 18 recording sessions and the outcome of this analysis was then averaged over all sessions, so as to account for variations of F(i) and Phase P(i) from one session to the next. For a precise estimation of phase difference between oscillations in attention information classification decoding and oscillations in sensory processing or behavioral outcome, a circular mean of the corresponding wavelet transform continuous phase difference between the two signals at frequency F(i) was extracted.

**Markov chain modeling of spatial attentional exploration**. Markov probabilistic chain models were used to characterize the spatial attention exploration strategy of each monkey from cue to target presentation. For each trial, (x,y) time series corresponding to the decoded spatial location of attention during the cue to target interval was collapsed onto the four possible screen quadrants, thus representing how attention moved from one quadrant to the other in time. Based on these discrete time series across all trials of a given session. A Markov chain model was used to estimate the probability that attention stayed in a given quadrant as well as the probabilities that it moved from the current quadrant to one of the three others. This model was performed using the Hmmestimate Matlab function of the Statistics and Machine Learning Toolbox. To compensate for possible idiosyncratic exploration biases of each monkey, the post-cue transition probabilities were normalized with respect to pre-cue spatial attention exploration transition probabilities. Transition probabilities were then normalized for each session and averaged over all sessions and both monkeys. This Markov chain modeling of

spatial attentional exploration strategy was independently performed for both tasks: the four cued-location and the two-cued location tasks.

**Reporting summary**. Further information on research design is available in the Nature Research Reporting Summary linked to this article.

## Data availability
The data that support the findings of this study are available from the corresponding author upon reasonable request. Data are still being analyzed for other purposes and cannot be made publically available at this time.

## Code availability
The code that supports the findings of this study is available from the corresponding author upon reasonable request. The code is still being used for other purposes and cannot be made publically available at this time.

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

## Acknowledgements

S.B.H was supported by ERC Brain3.0 #681978, ANR-11-BSV4-0011 & ANR-14-ASTR-0011-01, LABEX CORTEX funding (ANR-11-LABX-0042) from the Université de Lyon, within the program Investissements d'Avenir (ANR-11-IDEX-0007) operated by the French National Research Agency (ANR). R.V. was funded by ERC P-CYCLES #614244, ANR OSCIDEEP under the Grant agreement n°ANR-19-NEUC-0004 & ANR "Investing for the Future – PIA3" program n°ANR-19-PI3A-0004. We thank research engineer Serge Pinède for technical support and Jean-Luc Charieau and Fidji Francioly for animal care. All procedures were approved by the local animal care committee (C2EA42−13-02-0401-01) and the Ministry of research, in compliance with the European Community Council, Directive 2010/63/UE on Animal Care.

## Author contributions

Conceptualization, S.B.H. C.G. and R.V.; Data Acquisition, S.B.H.H., F.D.B.; Methodology, C.G., S.B.H., and Y.B.P.; Investigation, C.G., S.B.H.; Writing – Original Draft, S.B.H. and C.G.; Writing – Review & Editing, S.B.H., C.G., R.V.; Funding Acquisition, S.B.H.; Supervision, S.B.H.

## Competing interests

The authors declare no competing interests.
