## [Peer Review File · Nature Communications]

Reviewers' Comments:

Reviewer #1:

Remarks to the Author:

Following up on previous high-quality work from the lab, the authors use decoding methods to track the attentional focus of the animals, and find that the location is moving rhythmically in the alpha range. Furthermore, the authors show that a similar rhythm can also be found in the power spectrum of the raw spiking activity itself. They further show that this rhythm correlates with target and distractor decoding accuracy, and can predict hit and false alarm rates.

In particular the part on the rhythmic movement of the attentional focus is in my view very novel and exciting. It allows a reinterpretation of previous findings on rhythmic fluctuations of attention, the rhythm does not seem to correspond to 'hopping' between remote stimulus locations, but to a more local 'wobble' around a single stimulus location. Previous literature relied mostly on indirect behavioral or LFP/EEG/MEG recordings, while the current work employs large populations of single neurons in FEF (which is generally considered the source of spatial attention). The findings from the current work should therefore be taken seriously, in particular as findings from previous studies are largely replicated in the current work. Furthermore, the lack of evidence for a 'hopping' between distant stimuli is also in line with previous studies based on spiking activity recorded in monkey V4, where they find that low-frequency correlations between neurons are only present within a hemisphere, not between two recording sites in different hemispheres (where the two potential targets are overlapping the receptive fields of the two recording sites) (Cohen & Maunsell J. Neurosc. 2010; Cohen & Maunsell Neuron 2011).

A main question that I wonder about is what the relationship is between the recorded receptive fields (RFs) of the different channels and the oscillation. As the data is from two probes in the two different hemispheres, you can expect that the receptive fields are from largely separate RFs as well. It could be expected that the phase of the alpha rhythm in those two populations are different and maybe potentially even opposite. Still, the results on the average across site (the so-called 'super MUA') seem to suggest that there is a common phase across RFs. It would be relevant to understand the differences in results for different RFs and in particular the two different probes.

Are the results in Figures 3D, 4, 7 and 8A,B similar when taking the activity from one probe only, or sites with neighboring RFs? In particular, do the phase lags plotted as insets in panels 4B,D and 7B,D become less variable when selecting the channels corresponding to the RF of the relevant target/distractor? And does the power spectrum in the right panel in Figure 8A look similar when selecting nearby receptive fields?

Minor points

- The flow in the paper could be optimized. In my view, it would make more sense to start with the results in Figure 8, as this builds on top of the previous great work from the lab, and is the most novel finding. Furthermore, a number of times, referencing of the figures in the text is not in the expected order. For instance on line 134, Figure 5 is referenced before Figures 4C-E. This makes the text somewhat confusing to read. This also happens in the legends, in particular for Figure 2, where the insert in panel B belongs to panel A, more than panel B. I can understand that the authors tried to save space, but this solution does not help the readability of the text.

- Could you also show a similar plot as in Figure 2B for the average across sessions? Do you find two peaks, is there an indication of an additional peak in the theta range, as in Figure 5B, and maybe also 8A right panel?

- The referencing of the panels is not consistent in terms of whether upper or lower case is used. It seems Nature Comm. requires lower case. Also, the labels of the panels have a rather odd minus sign behind them. This also appears in the legend of Figure 4.

- On line 60, it seems that it should read ".. fig. 2A), fluctuations in ..". Similar, on line 131 "1.5), in contrast with" , and on line 196, "(fig.7), along the same"

- What do the red bars in panel 2D indicate?

- Please explain what you mean by 'super MUA' on line 91, maybe just use 'averaged MUA across channels' instead of it?
- On lines 95 and 96, reference is made to Figure 2 instead of Figures 3.
- There seems to be no data backing up the claim in lines 96-98.
- On line 127, the reference to "18" seems not in the correct format.
- In figure legend 3, line 104, it would be helpful to add "(B) raw (black trace) and alpha filtered single trial population super MUA calculated over the 48 MUA channels (grey trace)."
- Panels 4C,E and 7C,E miss labels on the y-axis, as they seem to be different from the labels in panels 4B,D and 7B,D.
- On line 261-262, the word 'either' is used twice, maybe remove one?
- The sentence on line 273 is confusing. Maybe write: "vary between the period before (fig. 9A.."
- It might help to use different color schemes for the two epochs in Figure 9B. At the moment they are hard to distinguish, and it is unclear what the numbers 1 and 2 refer to.
- In the discussion, for example at line 397, it might be relevant to discuss previous literature about the time scale of shifts of attention. For example Buschman & Miller Neuron 2009, who find a much faster rhythm.
- I'm missing the author contributions.

Timo van Kerkoerle

Reviewer #2:

Remarks to the Author:

The authors report that the brain explores visual space rhythmically in the alpha band via "attentional" saccades of the mind's eye. This exploration is entrained with alpha rhythms in the pre-frontal cortex. This rhythmic sampling was task-dependent (cued space was explored) and correlated with behavior. Further, a distractor had a greater impact on behavior if appeared at the peak of an alpha cycle.

This is a nice manuscript that fits squarely with, and expands upon, mounting evidence that "sustained" attention is actually rhythmic. One can always quibble about analyses (I do have a couple) but I think that the authors make a convincing case as is. My comments are meant to improve an already nice manuscript.

1. The use of a "Super MUA" is unusual. Most investigators have used local field potentials (LFPs) for such analyses. I think this MS would benefit from an explanation/justification for using the Super MUA. Were LFPs recorded? Did the Super MUA track the LFPs? Knowing whether or not the Super MUA and LFPs were more or less equivalent would be helpful in relating this work to other studies and in guiding future work.

2. In the behavioral task, why is it that only two opposing diagonals were used instead of all four quadrants for half of the sessions? This is not explained well.

3. The analyses focused on the peak power which was in the alpha band. Figure 5 suggests a theta band component as well. Were analyses run on a wider range of frequencies?

4. The data presented in Figure 6 is somewhat worrisome. A correlation being significant is not the whole story. What is also important is how much variance is explained. An R^2 of 0.25 is a poor explanation of variance. My eyeball suggests that dropping just 3 outlying data points (the two highest and one lowest on the Y-axis) would render the results non-significant. It seems that the data comes from 16 sessions (with two cued positions). What about the other 19 sessions (with four cued

positions)? More data would help. Or it may hurt. Why was less than half of the data used for this analysis?

5. Recordings were made with laminar electrodes. Was there any layer differences? A number of studies have reported that slower rhythms like alpha are strongest in deep layers of the cortex.

6. The authors cite a number of studies showing rhythmic attentional sampling fixed at one location. The implication is that this is the first study to show "attentional saccades" of the mind's eye. It is not. Buschman reported virtually the same result in the FEF (Buschman and Miller, 2009 Neuron). This, by the way, predates the other work cited by the authors but it is not cited despite the seemingly similar results. Buschman also found rhythmic covert search (attentional saccades of the mind's eye) that correlated with rhythms in the pre-frontal cortex. It was at a faster pace (18-24 Hz) but that is likely a trivial difference. The "clock" may run at different depending on task demands etc. I'm not saying that this precedent lessens the value of this work. The authors go beyond that prior work by showing effects of distractors, task demands etc. But the Buschman paper should not only be cited but also discussed given the similarity of the results. To not do so would make the current results seem more novel than they actually are. That is not good scholarship.

Reviewer #3:

Remarks to the Author:

The present manuscript, titled "Prefrontal attentional saccades explore space rhythmically," used multi-unit activity recorded in the prefrontal cortex (FEF) of macaques to track the locus of spatial attention during a cue-target delay. The results indicate, in line with other recent work, that spatial attention is a dynamic process, with sampling occurring rhythmically at a frequency in the range of 7-12 Hz. The authors use novel approaches to decode and track the attentional spotlight. Their results provide further support for a new, rhythmic characterization of spatial attention, and further evidence of the underlying neural basis. Below I provide some specific comments/questions:

(1) Generally speaking, I do not think that the Summary fully conveys the novel aspects of the present manuscript. See the following comments, 2-4.

(2) In the Summary, "Recent studies suggest that attention samples space rhythmically through oscillatory interactions in the frontoparietal network. However, the precise mechanism through which prefrontal cortex, at the source of attention control signals, organizes this rhythmic exploration of space remains unknown." This precise mechanism does not seem to be defined in the summary. How specifically does the present manuscript go beyond other recent studies (Spyropoulos et al., PNAS, 2018; Fiebelkorn et al., Neuron, 2018; Helfrich et al., Neuron, 2018; Kienitz et al., Current Biology, 2018; Fiebelkorn et al., Nature Communications, 2019) at getting at the underlying mechanism?

(3) In the Summary, I do not think the "the mind's eye" is an appropriate metaphor for covert attention. The "mind's eye" typically refers to mental imagery.

(4) In the Summary (and throughout the manuscript), "We propose that this rhythmic prefrontal attentional spotlight dynamics corresponds to a continuous overt exploration of space via alpha-clocked attentional saccades." I disagree with this characterization or maybe just the wording. First, the use of overt to refer to something that's happening covertly, when overt is typically used in this context to refer to exploratory movements. Similarly, attentional "saccades" is a misleading term, as "saccade" is defined as an eye movement. I think the authors need more clarity in their description, maybe "We propose that these rhythmic dynamics in prefrontal cortex reflect an alpha-clocked

sampling of the visual environment that continues to occur in the absence of eye movements.”

(5) Similar to other recent papers, the authors link rhythmic changes in neural activity with changes in sensory processing and perceptual outcomes. The authors describe rhythmic changes in neural activity as supporting either exploitation or exploration. A recent TICS paper, by Fiebelkorn and Kastner (2019) describes rhythmic changes in neural activity as supporting either sampling or shifting. The authors should discuss how their own ideas are similar to or contrast with this previous, related work.

(6) The authors used a behavioral task with 100% cue validity, which means there is no clear behavioral measure of attentional deployment. The authors do show that two different distractor types are associated with worse performance at the cued location. Here, I have two requests: first, the authors use two different types of distractors but do not differentiate between them when reporting how they influence behavioral performance (Figure 1B). Please report performance in the presence of distractors separately for the distractor types. Second, could the authors please provide some basic neurophysiological evidence of attentional deployment at the cued location? The authors could provide MUA during the cue-target interval when response fields overlapped the cued location relative to when response fields overlapped a non-cued location.

(7) To what extent does the averaging window for decoding influence the observed frequency of the attentional sampling? The 50-ms averaging window is equivalent to a 20-Hz sampling rate, which means the Nyquist frequency is 10 Hz. The authors should not report findings at any frequencies above 10 Hz, yet the figures (e.g., Figure 2B) show results from 0-55 Hz. The averaging window is acting as a low-pass filter at higher frequencies.

(8) “The decoder was trained... at a specific time during the cue-target delay,” please provide clarification here. What specific time? If the attentional spotlight is dynamic, then the specific time point relative to the cue should influence how the decoder is classifying the locus of attention (i.e., did that time point fall at a peak or a trough in the sampling rhythm). The primary point of the paper is that the attentional spotlight is highly dynamic, shifting around the visual scene, but shouldn't that make it difficult then to train the decoder in the first place?

(9) The sampling frequency observed in the present results is a bit higher than the sampling frequency described in some of the previous work. Might this somewhat higher sampling frequency (at ~9 Hz) reflect the specific experimental design, i.e., 100% cue validity? Such conditions might influence how/whether the attentional spotlight is split across multiple locations (see Landau and Fries, *Current Biology*, 2012 and Re et al., *Current Biology*, 2019).

(10) In the results, “Oscillations in the attention-related population activity can either reflect a global rhythmic entrainment of the entire FEF population or changes in the FEF population code at a specific frequency.” Would the authors unpack this a bit, I didn't understand the difference between these possibilities until I read further. Also please define “super MUA” in the main text of the paper (it is already defined in the methods). Apologies if I missed the definition in the main text.

(11) It is very difficult to see the alpha-filtered “super MUA” in Figure 3B.

(12) The authors demonstrate evidence of oscillatory patterns in the behavioral data, with peaks in both the theta and alpha ranges. Is there any evidence of a relationship between theta and alpha in the neural data (e.g., Fiebelkorn et al., *Nature Communications*, 2019)?

(13) Did the authors measure whether changes in MUA are linked to the phase of oscillatory activity in the local field potentials?

We would like to thank both the editors and the reviewers for their positive appreciation of our work and the very constructive comments they have provided us with. We feel our manuscript has been clarified and strengthened by these comments. Below is a point by point response to the reviewers' questions. Due to the new analyses, several supplementary figures have been added and supplementary information has been added. A supplementary methods section & a supplementary reference section have also been added and reproduced at the end of the present document. All references in this response to the reviewers' comments are listed in the supplementary references.

Reviewer #1

Following up on previous high-quality work from the lab, the authors use decoding methods to track the attentional focus of the animals, and find that the location is moving rhythmically in the alpha range. Furthermore, the authors show that a similar rhythm can also be found in the power spectrum of the raw spiking activity itself. They further show that this rhythm correlates with target and distractor decoding accuracy, and can predict hit and false alarm rates.

In particular, the part on the rhythmic movement of the attentional focus is in my view very novel and exciting. It allows a reinterpretation of previous findings on rhythmic fluctuations of attention, the rhythm does not seem to correspond to 'hopping' between remote stimulus locations, but to a more local 'wobble' around a single stimulus location. Previous literature relied mostly on indirect behavioral or LFP/EEG/MEG recordings, while the current work employs large populations of single neurons in FEF (which is generally considered the source of spatial attention). The findings from the current work should therefore be taken seriously, in particular as findings from previous studies are largely replicated in the current work. Furthermore, the lack of evidence for a 'hopping' between distant stimuli is also in line with previous studies based on spiking activity recorded in monkey V4, where they find that low-frequency correlations between neurons are only present within a hemisphere, not between two recording sites in different hemispheres (where the two potential targets are overlapping the receptive fields of the two recording sites) (Cohen & Maunsell J. Neurosc. 2010; Cohen & Maunsell Neuron 2011).

We would like to thank Timo van Kerkoerle for his positive appreciation of our work and the very constructive comments he provided us with. Below is a point by point response to his comments.

1- A main question that I wonder about is what the relationship is between the recorded receptive fields (RFs) of the different channels and the oscillation. As the data is from two probes in the two different hemispheres, you can expect that the receptive fields are from largely separate RFs as well. It could be expected that the phase of the alpha rhythm in those two populations are different and maybe potentially even opposite. Still, the results on the average across site (the so-called 'super MUA') seem to suggest that there is a common phase across RFs. It would be relevant to understand the differences in results for different RFs and in particular the two different probes. Are the results in Figures 3D, 4, 7 and 8A,B similar when taking the activity from one probe only, or sites with neighboring RFs? In particular, do the phase lags plotted as insets in panels 4B,D and 7B,D become less variable when selecting the channels corresponding to the RF of the relevant target/distractor? And does the power spectrum in the right panel in Figure 8A look similar when selecting nearby receptive fields?

This is a very relevant point. In the following, we first provide information of the neuronal population response properties, we then compare frequency and phase of attentional rhythm across hemispheres.

PART 1. The following information is now added to the supplementary information:

"Note 1: Description of the neuronal population response properties"

The recorded receptive fields are quite large, as typically described in the FEF. Sixty-one percent of the MUA channels had a significant **target related** response on correct trials. Of these, 23.5% of the recorded RFs encompass one visual quadrant, 24.2% encompass two ipsilateral visual quadrants, 4.6% encompass two opposing visual quadrants, 21.6% encompass three visual quadrants and 26.1% encompass 4 visual quadrants.

Seventy-three percent of the MUA channels had a significant **attention related** response on correct trials. Of these, 14.1% of the recorded RFs encompass one visual quadrant, 14.4% encompass two ipsilateral visual quadrants, 3.6% encompass two opposing visual quadrants, 20.2% encompass three visual quadrants and 47.7% encompass 4 visual quadrants.

This diverse receptive field structure of the data was critical for the success of the linear decoding approach that we are using here. Noteworthy is the fact that, in addition to significantly modulated neurons, non-significantly modulated neurons also contributed to the decoder (Farbod Kia et al. 2011).

Fig. S2 further reports the MUA spatial attention selectivity on an exemplar MUA signal, an exemplar session and across recording sessions.”

Fig. S2 and its legend are reproduced below.

Figure S2: **MUA spatial attention selectivity.** (a) Single MUA mean (\pm s.e.), when cue is orienting attention towards the preferred (black) or the anti-preferred (gray) spatial location, during the cue to target interval. X-axis represents time around the cue to target interval. (b) MUA spatial attention selectivity for a representative recording session. X-axis represents time around the cue to target interval. Y-axis represents individual channels, separated in left and right hemisphere channels. Each line represents, for each individual channel, the difference between the normalized neuronal response to a cue orienting attention towards the preferred spatial location and the normalized neuronal response to a cue orienting attention towards the anti-preferred

spatial location. White ticks represent the onset of statistically significant differences between these two signals (Wilcoxon, $p < 0.05$). (c) Distribution of a spatial attention index (Preferred-AntiPreferred)/(Preferred+AntiPreferred), computed over [-200 0] ms before target onset) across all MUA of all sessions. Red histogram corresponds to channels in which the neuronal activity during this time interval was significantly different between the preferred and the anti-preferred spatial attention responses (Wilcoxon, $p < 0.05$, gray, no significant difference).

This is also referenced in the main text as follows: "In order to access FEF attentional content in time, monkeys performed a manual response cued target-detection task (fig. 1a) while we recorded the MUA bilaterally from their FEF neuronal ensembles, using two 24-contacts recording probes (fig. 1c). Distractors were presented during the cue-to-target interval and target luminosity was adjusted so as to make the task difficult to perform without orienting attention (fig. 1b, fig. S1 independently reports the behavioral performance for both distractors at the landmarks and distractors in the workspace, fig. S2 and supplementary Note 1 report the MUA spatial attention selectivity across recording sessions)."

PART 2. As expected by reviewer #1, super MUAs computed across left and right electrodes on individual trials oscillated at a common rhythm as well as at a common phase. In contrast, while decoded population attention information also oscillated at a common rhythm when computed independently on the left and right probes, these signals were in anti-phase one with respect to the other.

This information is now added to the manuscript as supplementary fig. S6, and in the supplementary material as follows:

"Note 3: Frequency & phase of attentional rhythm across hemispheres.

We show that the spiking probability of individual MUA channels is modulated at an alpha-rhythm (fig. 3acd). A very clear alpha-oscillation can also be identified, when the activity of MUAs across all channels is computed for individual trials (super MUA, fig. 3b). This strongly suggests that MUAs from both hemispheres are phase locked. Confirming this prediction, fig. S6a shows, for an exemplar trial, the super MUA independently computed on the left and right MUA recordings. On this specific trial, both super MUAs oscillated at exactly the same rhythm and are perfectly in-phase. This is confirmed at the level of all trials and all sessions (peak frequency for left mean \pm s.e.=9.72Hz \pm 0.49Hz and right super MUAs mean \pm s.e.=9.8Hz \pm 0.56, Wilcoxon test for equal means, $p=0.45$; fig. S6b, across session distribution of phase differences between the left and right super MUA alpha content, circular mean=0.07°, circular s.d.=3.26°). This thus indicates that the spiking rate of the entire FEF population is modulated at a unique rhythm, in-phase between both hemispheres.

As discussed in the main text, super MUA rhythmicity reflects an alpha-clocked change in the FEF attention-related information, rather than an information-independent global alpha-modulation. Under the hypothesis that attention is mainly directed to either one hemisphere or the other (at least in the context of the current task), one expects the strict phase locking between the left and right super MUAs to translate into anti-phased left and right attention population contents. In other words, one expects that when attention information content is high in the right cortical population, it is at the same time low in the left cortical population and vice versa. Our current decoding is based on 48-channels. Repeating the cross-temporal decoding analyses presented in fig. 2 using one probe at a time, significantly decreases overall decoding performance (average classification performance of attention allocation: M1: bilateral 43.8%, unilateral 38.1%; M2: bilateral 39.5%, unilateral 33%, absolute chance level at 25%), all the more that our decoding is being performed at a high temporal resolution (50ms). In spite of this drop in overall decoding accuracy, peak oscillatory frequencies identified on the left and right neuronal population decoded attentional signal are indistinguishable (fig. S6c, $p=0.75$, Wilcoxon non-parametric test) and similar to the frequencies identified based on bilateral decoding (Wilcoxon non-parametric test, bilateral vs. left, $p=0.33$; bilateral vs. right, $p=0.20$). Importantly, the oscillations in the attentional signal between these two left/right neuronal populations are close to anti-phase (fig. S6d, circular mean=178.28°, circular s.e.=16.24°). Overall, this suggests an interhemispheric coordination in how the attentional spotlight explores visual space between the two hemispheres. This will need to be further explored.

Given that the attentional rhythms are indistinguishable whether identified on the left probe, the right probe or both probes, the analyses presented in fig. 4, 7 and 8 remain unchanged by this factor. Indeed, these analyses are uniquely driven by the identified attentional rhythms.”

Figure S6 is reproduced below:

Figure S6: **Frequency and phase of attentional rhythm across hemispheres.** (a) Single trial example of super MUA across left (grey) and right hemifield (black) recordings, on a representative session. (b) Distribution of phase relationship between the left and right super MUA alpha content, over all trials of all sessions (gray) and corresponding 95%CI (red). (c) Peak frequencies identified in the cross-temporal classification based on either right or left population information and (d) distribution of corresponding phase relationship (95%CI red).

This is also referenced in the main text as follows:

“Overall, inter-individual and inter-session variability was low and PFC attention-related information oscillated, in monkey M1 (resp. M2), at an average frequency of 9 Hz (fig. 2c, resp. 8.6 Hz). A clear phase-locking between these attention-related oscillations and cue onset can be seen across both monkeys (fig. 2d, M1: -75°; M2 -65°). This rhythmic oscillation of the PFC attentional spotlight is phase reset by cue presentation and actually pre-exists to cue presentation (see below). [...] Importantly, oscillations can similarly be identified from unilateral cortical recordings, in the same frequency range (fig. S6c). These oscillations are however in anti-phase between the left and right hemispheres (fig. S6d), suggesting an active inter-hemispheric coordination mechanism (suppl. Note 3).”

“However, this alpha rhythmic modulation of spiking probability does not reflect a global entrainment of the entire population. [...] Rather, the channels with highest change in normalized spiking activity change from one super MUA alpha peak to the next, thus reflecting a change in the FEF population code. These variations correspond to changes in the spatial allocation of the attentional spotlight that will be described hereafter. Super MUAs independently computed across left and right electrodes on individual trials oscillated at a common rhythm (fig. S6a) as well as at a common phase (fig. S6b, suppl. Note 3). This contrasts with the fact that decoded population attention information also oscillated at a common rhythm (fig. S6c) when computed

independently on the left and right probes, but in anti-phase one with respect to the other (fig. S6d). This confirms that these variations in MUA spiking probability correspond to changes in the spatial allocation of the attentional spotlight and suggest an active inter-hemispheric coupling mechanism.”

2- In the discussion, for example at line 397, it might be relevant to discuss previous literature about the time scale of shifts of attention. For example Buschman & Miller Neuron 2009, who find a much faster rhythm.

The following text has been added to the introduction: “Recent works propose that neural oscillations in the fronto-parietal network organize alternating attentional states or shifts in attention that in turn modulate perceptual sensitivity (Fiebelkorn et al., 2018, 2019, Buschman & Miller, 2009).”

This work is also discussed as follows in the discussion section:

“Attentional periodicity in monkey FEF has been suggested before (Buschman & Miller, 2009), albeit with a faster frequency (18-25Hz). This frequency difference may reflect disparities in experimental design or task difficulty. Alternatively, it could reflect specific differences between FEF LFP and MUA processes (fig. S8a). The present study goes beyond this prior work by explicitly decoding the position of attention over time, and exploring the effect of distractors and task contingencies.”

It is also cited in the result section as follows:

“Previous studies indicate a coupling between LFPs theta frequency content and behavior (Fiebelkorn et al., 2018) as well as between LFP beta frequency content and spiking activity and behavior (Buschman and Miller, 2009). An important question is thus whether changes in super MUAs are linked to the phase of oscillatory activity in the local field potentials. A significant alpha component is present in the super MUAs (fig. S8b) but not in the LFPs (fig. S8a). However, a significant phase-phase coupling can be identified between these two signals, in the alpha range (7-12Hz) as well as in the beta frequency range (18-30Hz, fig. S8d, suppl. Note 5). The functional significance of this coupling, its directionality and its causal relationship to attention and perception remains to be explored.”

3- Could you also show a similar plot as in Figure 2B for the average across sessions? Do you find two peaks, is there an indication of an additional peak in the theta range, as in Figure 5B, and maybe also 8A right panel?

This following text is now added to the result section as is, following the discussion of fig. 2b.

“Fig. S5 represents the average normalized power spectrum of spatial attention classification (same time window as in fig. 2b) for all sessions. Two significant local maxima can be identified against the 95% confidence interval, one in the theta range and one in the alpha range. The alpha peak is higher than the theta peak and coincides with the attentional rhythms discussed in the present work. Importantly, while the alpha attention information oscillation can be identified in all sessions (100%), the theta oscillation can only be identified in 68% of the sessions (13/19 sessions, significance assessed against the 95% confidence interval). In the following, we focus on the higher and most significant frequency peak, namely the alpha power.”

Reference to this fig. S5 is also added to the discussion of hit rate rhythmic variations as follows:

“... two significant oscillatory peaks are observed onto behavior, one in the theta (3 to 5 Hz) frequency band, and one in the alpha (9 to 14 Hz) frequency band (fig. 5B), thus reproducing previous behavioral observations^{20–24,37}. These two peaks coincide with those identified in the prefrontal attention-related information (fig. S5), as well as with those identified in the FEF LFPs (Fiebelkorn et al., 2018). This alpha peak expresses in a frequency range that is lower than the FEF-theta locked alpha peak identified in the pulvinar (Fiebelkorn et al., 2019).”

Figure S5 is replicated below:

Figure S5: *Averaged peak oscillations of prefrontal attention-related information across all sessions.* Normalized power identified in the cross-temporal classification reference interval (fig. 2b), (dark grey: significant frequencies relative to 95% C.I.) for all sessions (4-cued task version).

Minor points

1- The flow in the paper could be optimized. In my view, it would make more sense to start with the results in Figure 8, as this builds on top of the previous great work from the lab, and is the most novel finding.

We do agree with reviewer 1 that fig. 8 is indeed an important addition to the current understanding of spatial attention. However, from our experience in presenting the data, the methodological aspects of the related analysis are difficult to convey and relate with the current literature without going through the analyses presented in the previous figures. We thus prefer to keep the organization of the paper in its current form.

Furthermore, a number of times, referencing of the figures in the text is not in the expected order. For instance, on line 134, Figure 5 is referenced before Figures 4C-E. This makes the text somewhat confusing to read.

This early reference to fig. 5 has now been removed.

This also happens in the legends, in particular for Figure 2, where the insert in panel B belongs to panel A, more than panel B. I can understand that the authors tried to save space, but this solution does not help the readability of the text.

Fig. 2 has been modified such that the previous inset to fig. 2b is now an independent subplot of fig. 2. Legend and main text have been changed accordingly. Fig. 2 is duplicated below.

Figure 2: **Oscillation of prefrontal attention-related information.** (a) Cross-temporal classification around cue presentation ([-100 : 1200 ms], step of 10ms, averaging window of 50ms) for an exemplar session. White contour: 95% confidence interval as assessed from trial identity random permutation. Black contour: close-up of the cross-temporal classification ([testing time: 500-1150 ms] post-cue; training time: [400-575 ms]) and corresponding mean classification along testing time (black). (b) Normalized power in this cross-temporal classification interval, (red line: 95% confidence interval) for the exemplar session presented in (a). (c) Average +/- s.d. of peak power in a 7-12 Hz interval, over all sessions, for each monkey (M1: black, M2: gray), in the 4-cued locations task version. (d) Circular distribution of signal phase with respect to cue onset (red bars), at identified peak frequency (mean phase: M1: black, -75°, M2: gray, -65°).

The referencing of the panels is not consistent in term of whether upper or lower case is used. It seems Nature Comm. requires lower case. Also, the labels of the panels have a rather odd minus sign behind them. This also appears in the legend of Figure 4.

This has now been homogenized.

2- On line 60, it seems that it should read "... fig. 2A), fluctuations in ...". Similar, on line 131 "1.5), in contrast with", and on line 196, "(fig.7), along the same"

All these have been corrected – thank you for this thorough feedback!

3- What do the red bars in panel 2D indicate?

This has now been clarified in the legend as follows: "(d) Circular distribution of signal phase with respect to cue onset (red bars), at identified peak frequency (mean phase: M1: black, -75°, M2: gray, -65°)."

4- Please explain what you mean by 'super MUA' on line 91, maybe just use 'averaged MUA across channels' instead of it?

We now define what we mean by super MUA not only in the methods but also in the main text. We choose to keep the term super MUA throughout the manuscript as the use of 'averaged MUA across channels' makes the reading more difficult. The text has now been changed as follows. Text in between brackets has been added in response to comment 10 of reviewer 3: 'Fig. 3A represents, for one exemplar recording probe, on an exemplar trial, and for each recording channel, the time epochs at which spiking-rate exceeds the 65% of the maximum spiking regime of the individual channel. [On every single channel, these high spiking probability epochs do not appear to follow a systematic rhythm, thus contradicting a global rhythmic entrainment hypothesis. Rather, this high spiking probability organizes in discrete epochs, distributed over all recording channels. The

hypothesis that changes in the FEF population code (and thus high spiking probability epochs) take place at a specific frequency implies that average MUA over all channels on a given trial will show marked rhythmic variations in firing rate in time. Fig. 3b confirms this hypothesis.] For this individual trial, a *super MUA* signal was computed by averaging the spiking activity of the 48 recording channels on this specific trial. Peaks of alpha oscillations are clearly identified on the super MUA of the same individual trial (fig. 3b,) and plotted against the spiking probability changes represented in fig. 3a.'

5- On lines 95 and 96, reference is made to Figure 2 instead of Figures 3.

This has now been corrected.

6- There seems to be no data backing up the claim in lines 96-98.

This has now been further clarified in the text as follows: "The high spiking probability epochs of individual channels coincide with peak alpha oscillatory phases in the super MUA. This is captured by a spectral analysis of changes in spiking probability in a frequency range running from 5 to 15 Hz (see material & methods). Most channels of fig. 3a display a modulation of spiking probability in an 8 - 12 Hz frequency range (fig. 3c, color code matching fig. 3a). This holds true for all sessions (fig. 3d, mean+/-s.e.). However, this alpha rhythmic modulation of spiking probability does not reflect a global entrainment of the entire population. This can be seen in fig. 3a in which individual channels do not exhibit on any given trial, high spiking rate synchronously at each identified alpha cycle. This is also captured in fig. 3c, as the degree of alpha locking of spiking activity varies from one channel to the next. Rather, the channels with highest change in normalized spiking activity change from one super MUA alpha peak to the next, thus reflecting a change in the FEF population code. These variations correspond to changes in the spatial allocation of the attentional spotlight that will be described hereafter."

7- On line 127, the reference to "18" seems not in the correct format.

This has now been corrected.

8- In figure legend 3, line 104, it would be helpful to add "(B) raw (black trace) and alpha filtered single trial population super MUA calculated over the 48 MUA channels (grey trace)."

This has now been corrected.

9- Panels 4C,E and 7C,E miss labels on the y-axis, as they seem to be different from the labels in panels 4B,D and 7B,D.

y-axis labels have now been added.

10- On line 261-262, the word 'either' is used twice, maybe remove one?

This has now been corrected.

11- The sentence on line 273 is confusing. Maybe write: "vary between the period before (fig. 9A..)"

This has now been corrected.

12- It might help to use different color schemes for the two epochs in Figure 9B. At the moment they are hard to distinguish, and it is unclear what the numbers 1 and 2 refer to.

Figure 9 and its legend have been changed as suggested by reviewer 1.

Figure 9: **Attentional spotlight exploration is an endogenous process affected by task events.** (a) Distribution of amplitude of attentional displacement between one PFC attentional position and the next, in the pre-cue period (black) and in the cue-to-target interval (gray). PreCue (b) and postCue (c) heat maps of the spatial distribution of the decoded attentional spotlight.

13- I'm missing the author contributions.

Author contribution has now been added.

Reviewer #2 (Remarks to the Author):

The authors report that the brain explores visual space rhythmically in the alpha band via “attentional” saccades of the mind’s eye. This exploration is entrained with alpha rhythms in the pre-frontal cortex. This rhythmic sampling was task-dependent (cued space was explored) and correlated with behavior. Further, a distractor had a greater impact on behavior if appeared at the peak of an alpha cycle.

This is a nice manuscript that fits squarely with, and expands upon, mounting evidence that “sustained” attention is actually rhythmic. One can always quibble about analyses (I do have a couple) but I think that the authors make a convincing case as is. My comments are meant to improve an already nice manuscript.

We would like to thank reviewer #2 for his/her positive appreciation of our work and the very constructive comments he/she has provided us with. Below is a point by point response to his/her comments. Due to the new analyses, several figures have been renumbered.

1. The use of a “Super MUA” is unusual. Most investigators have used local field potentials (LFPs) for such analyses. I think this MS would benefit from an explanation/justification for using the Super MUA. Were LFPs recorded? Did the Super MUA track the LFPs? Knowing whether or not the Super MUA and LFPs were more or less equivalent would be helpful in relating this work to other studies and in guiding future work.

Following suggestion of reviewers #1 & #3, we also redefine “super MUA” in the result section –but we choose to keep this nomenclature rather than averaged MUA across channels. We now also provide a more extensive justification of using the super MUA measure in response to the comments of reviewer #2. We also rephrase the beginning of this paragraph to unpack what we mean by this sentence, following the request of reviewer #3. This paragraph now reads as follows: “Oscillations in the attention-related population activity can either reflect a global rhythmic entrainment of the entire FEF population (i.e. all spiking rates of all neurons throughout the FEF, changing their firing rates synchronously, being enhanced or suppressed coherently) or changes in the FEF population code at a specific frequency (i.e. the spiking rates of only some FEF neurons changing their firing rates, being enhanced or suppressed, at any peak or trough of the identified oscillations, each specific neuronal combination corresponding to a specific spatial attentional code). Fig. 3a represents, for one exemplar recording probe, on an exemplar trial, and for each recording channel, the time epochs at which spiking-rate exceeds the 65% of the maximum spiking regime of the individual channel. On every single channel, these high spiking probability epochs do not appear to follow a systematic rhythm, thus contradicting

a global rhythmic entrainment hypothesis. Rather, this high spiking probability organizes in discrete epochs, distributed over all recording channels. The hypothesis that changes in the FEF population code (and thus high spiking probability epochs) take place at a specific frequency implies that average MUA over all channels on a given trial will show marked rhythmic variations in firing rate in time. Fig. 3b confirms this hypothesis. For this individual trial, a *super MUA* signal was computed by averaging the spiking activity of the 48 recording channels. Peaks of alpha oscillations are clearly identified on the super MUA of the same individual trial (fig. 3b,) and plotted against the spiking probability changes represented in fig. 3a. The high spiking probability epochs of individual channels coincide with peak alpha oscillatory phases in the super MUA. This is captured by a spectral analysis of changes in spiking probability in a frequency range running from 5 to 15 Hz (see material & methods). Most channels of fig. 3a display a modulation of spiking probability in an 8 - 12 Hz frequency range (fig. 2c, color code matching fig. 2a). This holds true for all sessions (fig. d, mean+/-s.e.). However, this alpha rhythmic modulation of spiking probability does not reflect a global entrainment of the entire population. [...] Rather, the channels with highest change in normalized spiking activity change from one super MUA alpha peak to the next, thus reflecting a change in the FEF population code. These variations correspond to changes in the spatial allocation of the attentional spotlight that will be described hereafter.”

Reviewer #2's questions on how super MUA activity track LFP properties is an excellent question that would allow to relate our current observations to previous literature based on correlational analyses between behavior and LFPs. This is now presented in a new supplementary information section as follows:

“Note 5: Phase relationship between super MUA and LFPs

The super MUAs reflect the general alpha-clocking of the entire bilateral FEF neuronal population. This activity is phase locked between the two hemispheres (fig. S6 and related supplementary information). In contrast, individual LFPs reflect both long range inter-areal and local processes on each session. Fig. S8 represents the frequency content (mean +/- s.e.) of both the task-related LFPs (fig. S8ab) and the super MUAs (fig. S8c) across all sessions. LFPs show an enhanced frequency content in the lower theta range (3-5Hz) as well as in the beta frequency range (18-30Hz). These observations are consistent with previous reports (Fiebelkorn et al., 2018, Buschman and Miller, 2009). In contrast, while the theta peak of the super MUAs is weak, these signals show a marked frequency peak in the alpha range (7-12Hz, coinciding with the frequency range described in the present work, fig. 2 & 3), as well as a consistent peak in the beta frequency range (18-30Hz). Alpha oscillatory mechanisms thus appear to be specific to the super MUAs. Fig. S8d represents phase-phase coherence between the LFP and super MUA signals. Coherence is enhanced in the three frequency bands identified in fig. S8ab, namely the lower theta range (3-5Hz), the alpha range (7-12Hz), as well as in the beta frequency range (18-30Hz). Overall, this thus suggests a strong phase coupling between the FEF LFPs and super MUAs. The functional significance of this coupling, its directionality and its causal relationship to attention and perception remains to be explored.”

This section is now referenced in the main result section as follows:

“Rather, the channels with highest change in normalized spiking activity change from one super MUA alpha peak to the next, thus reflecting a change in the FEF population code. These variations correspond to changes in the spatial allocation of the attentional spotlight that will be described hereafter. Super MUAs independently computed across left and right electrodes on individual trials oscillated at a common rhythm (fig. S6a) as well as at a common phase (fig. S6b, suppl. Note 5). This contrasts with the fact that decoded population attention information also oscillated at a common rhythm (fig. S6c) when computed independently on the left and right probes, but in anti-phase one with respect to the other (fig. S6d). This confirms that these variations in MUA spiking probability correspond to changes in the spatial allocation of the attentional spotlight and suggest an active inter-hemispheric coupling mechanism.

[...]

Previous studies indicate a coupling between LFPs theta frequency content and behavior (Fiebelkorn et al., 2018) as well as between LFP beta frequency content and spiking activity and behavior (Buschman and Miller, 2009). An important question is thus whether changes in super MUAs are linked to the phase of oscillatory

activity in the local field potentials. A significant alpha component is present in the super MUAs (fig. S8b) but not in the LFPs (fig. S8a). However, a significant phase-phase coupling can be identified between these two signals, in the alpha range (7-12Hz) as well as in the beta frequency range (18-30Hz, fig. S8d, suppl. Note 5). The functional significance of this coupling, its directionality and its causal relationship to attention and perception remains to be explored.”

Figure S8 is replicated below:

Figure S8: **Alpha phase coherence between super MUAs and LFPs.** (a) Averaged LFP power spectrum (mean +/- s.e across all channels and sessions, dark grey: significance w/ 95%CI). (b) Close-up on low-frequency inset in (a), after 1/f correction. (c) Averaged super MUA power spectrum (mean +/- s.e, across all channels and sessions, dark grey: significance w/ 95%CI). (d) Averaged phase coherence between super MUAs and LFPs, calculated using pairwise phase consistency (PPC) (mean +/- s.e, across all channels and sessions, dark grey: significance w/ 95%CI).

2. In the behavioral task, why is it that only two opposing diagonals were used instead of all four quadrants for half of the sessions? This is not explained well.

We are sorry that this was not clear enough in the original version of the manuscript. This is now clarified as follows in the methods section: “Overall, data was collected for 19 sessions (M1 10 Sessions, M2 9 Sessions). The behavioral performance of each animal is presented in fig. 1b, for hit, miss and false alarm trials. In order to characterize whether the attentional temporal dynamics and attentional exploration trajectories described in this study were influenced by task structure, a second two-position variant of the above described task was also presented to the monkey. In this task, while the four landmarks were present all throughout the task as previously, only two diagonally opposite positions amongst the four were cued all throughout the session. The pair of cued stimuli changed from one session to the next. 16 such sessions were recorded (8 sessions for M1, 8 sessions for M2). All else was as described for the main four position task.”

The main text was changed as follows: “We show that attentional exploration trajectories depend on trial structure. The next question is thus whether the attentional temporal dynamics and attentional exploration trajectories described up to now are also influenced by task structure. To explore the influence of task contingencies onto the spatial deployment of rhythmic attention, we used Markov chain probabilistic

modelling to describe how the decoded PFC attentional spotlight explores space in two different versions of a cued target detection task, that only differed in the number and localization of the task relevant items: a first version (the one used up to now), in which the cue could orient attention to one of the four possible quadrants (18 sessions), and a second version in which the cue oriented attention to only two possible quadrants, placed along the diagonal one with respect to the other (16 sessions).”

3. The analyses focused on the peak power which was in the alpha band. Figure 5 suggests a theta band component as well. Were analyses run on a wider range of frequencies?

This following text is now added to the result section as is, following the discussion of fig. 2b.

“Fig. S5 represents the average normalized power spectrum of spatial attention classification (same time window as in fig. 2b) for all sessions. Two significant local maxima can be identified against the 95% confidence interval, one in the theta range and one in the alpha range. The alpha peak is higher than the theta peak and coincides with the attentional rhythms discussed in the present work. Importantly, while the alpha attention information oscillation can be identified in all sessions (100%), the theta oscillation can only be identified in 68% of the sessions (13/19 sessions, significance assessed against the 95% confidence interval). In the following, we focus on the higher and most significant frequency peak, namely the alpha power.”

Reference to this figure S5 is also added to the discussion of hit rate rhythmic variations as follows:

“... two significant oscillatory peaks are observed onto behavior, one in the theta (3 to 5 Hz) frequency band, and one in the alpha (9 to 14 Hz) frequency band (fig. 5B), thus reproducing previous behavioral observations ^{20–24,37}. These two peaks coincide with those identified in the prefrontal attention-related information (fig. S5), as well as with those identified in the FEF LFPs (Fiebelkorn et al., 2018). This alpha peak expresses in a frequency range that is lower than the FEF-theta locked alpha peak identified in the pulvinar (Fiebelkorn et al., 2019).”

Figure S5 is replicated below:

Figure S5: *Averaged peak oscillations of prefrontal attention-related information across all sessions.* Normalized power identified in the cross-temporal classification reference interval (fig. 2b), (dark grey: significant frequencies relative to 95% C.I.) for all sessions (4-cued task version).

4. The data presented in Figure 6 is somewhat worrisome. A correlation being significant is not the whole story. What is also important is how much variance is explained. An R2 of 0.25 is a poor explanation of variance. My eyeball suggests that dropping just 3 outlying data points (the two highest and one lowest on the Y-axis) would render the results non-significant. It seems that the data comes from 16 sessions (with two cued positions).

What about the other 19 sessions (with four cued positions)? More data would help. Or it may hurt. Why was less than half of the data used for this analysis?

Fig. 6 was initially carried on 19 sessions run with the 4 position task. Adding the 16 other sessions run with the 2 position task (which we apologize for not having done straight away) did not improve the correlation pattern probably due to inter-session high RT variability. We have thus decided to remove this analysis.

The aim of this analyses was to characterize the relationship between alpha oscillations in target encoding strength, alpha oscillations in hit rates and reaction time production. We have now simplified our approach, and we show that, when cumulating across all sessions, mean reaction times significantly vary at a marked alpha rhythm (fig. 6bc) mirroring the rhythmic variations in behavioral performance (fig. 5B). This section has now been reframed as follow:

“At a closer look, and as reported above for target processing, phase lag between signal and optimal target detection was quite variable from one session to the next (fig. 4d, inset). An important question is thus whether, on correctly detected targets, the above described alpha rhythmicity in target processing also impacts reaction time distribution. Although phase-lag between optimal target processing and optimal target detection (fig. 6a) was also variable across sessions (fig. 6b), mean reaction times, when cumulated over all sessions, significantly varied at a marked alpha rhythm (fig. 6c and 6d, please note that no theta rhythm can be identified). In other words, mirroring hit rates, reaction times were also modulated by attentional prefrontal alpha rhythm, suggesting that this rhythm contributed both to an enhanced perception as well as to speeded up responses, both processes being probably tightly coupled.”

New figure 6 is duplicated below:

Figure 6: Phase lag between optimal target encoding and optimal target detection behavioral response (a) presents a session to session variability (b). (c) Changes in reaction times to target presentation as a function of

time from cue presentation (detrended geometric mean \pm s.e). (d) Reaction time series power spectrum (mean \pm s.e., dark gray shaded areas, significance w/ 95%CI).

5. Recordings were made with laminar electrodes. Was there any layer differences? A number of studies have reported that slower rhythms like alpha like alpha are strongest in deep layers of the cortex.

This is a very interesting question! We thank reviewer #2 for pointing in this direction!

This information has now been added as is to the supplementary information as follows:

“Note 4: Alpha rhythm in superficial and deeper cortical layers

In our own data, as recordings were performed tangentially to FEF cortical surface, we have no direct assignment of the recorded MUAs to either superficial or deep cortical layers. However, previous studies have shown that pure visual neurons are predominantly located in the supragranular layers of the FEF while visuo-motor neurons are predominantly located in its infragranular layers (Bruce and Goldberg, 1985; Schall, 1991; Schall et al., 1995; Schall and Hanes, 1993; Schall and Thompson, 1999; Segraves and Goldberg, 1987). Pouget et al., (2009) further show that supragranular FEF neurons predominantly project to striate visual cortex while infragranular FEF neurons predominantly project to the superior colliculus (Fries, 1984; Leichnetz and Goldberg, 1988; Sommer and Wurtz, 2000). Interestingly, Buffalo et al. have shown that, in extra-striate area V4, the ratio between the alpha and gamma spike field coherence discriminate between LFP signals in deep (low alpha / gamma spike field coherence ratio) and superficial cortical layers (high alpha / gamma spike field coherence ratio, Buffalo et al., 2011a). In Ben hadj Hassen et al. (2019), we show that the LFP alpha / gamma spike field coherence ratio provides a very reliable segregation of visual and visuo-motor FEF MUAs at the same recording site. We thus consider that, as has been described for area V4, this LFP alpha / gamma spike field coherence allows for a reliable delineation of superficial and deep layers in area FEF, approximatively defined by a slope of $y=1.4x$. In the present study, we did not characterize the oculo-motor properties of the recorded MUA channels. However, these recordings were performed in exactly the same recording sites as in Ben Hadj Hassen et al. (2019). In the following (fig. S7a), we thus characterize the alpha / gamma spike field coherence ratio for all of our task-related LFP and segregate them as superficial or deep recording sites, using the same delineation slope as characterized in Ben hadj Hassen et al. (2019, see supplementary methods).

We then used the same methodology as presented in fig. 3, to quantify the frequency of high spiking probability epochs in superficial and deep FEF layers. Fig. S7b shows, for a representative trial, alpha filtered super MUA over superficial (red) and deep (blue) channels where alpha frequency amplitude appears higher in the deep (blue) contact. This is confirmed by a more global analysis (fig. S7c), in which we demonstrate that, across all channels and all sessions high spiking probability epochs expressed a rhythmicity in a strictly comparable alpha frequency range. Importantly, global MUA spiking alpha probability is significantly stronger in the deeper as compared to the superficial cortical layers (wilcoxon test $p<0.01^{**}$ from 7.5 to 12Hz), possibly suggesting that the alpha clock originates in the deeper cortical layers as reported in numerous studies and various cortical regions (Buffalo et al., 2011; Spaak et al., 2012).”

Figure S7: **Alpha spiking rhythm changes across prefrontal cortical layer** (a) Spike Field Coherence based layer segregation defined by LFP alpha / gamma ratio (slope $y=1.4x$). (b) Alpha filtered super MUA activity across deep (blue) and superficial (red) recorded contacts for a representative trial. (c) Deep layer MUA contact present significantly higher alpha locking spiking activity.

This information is also referenced in the main text as follows: “However, this alpha rhythmic modulation of spiking probability does not reflect a global entrainment of the entire population. Rather, the channels with highest change in normalized spiking activity change from one super MUA alpha peak to the next, thus reflecting a change in the FEF population code. These variations correspond to changes in the spatial allocation of the attentional spotlight that will be described hereafter. [...] Importantly, the identified alpha clocking frequency range didn’t vary between superficial and deep cortical layers, indicating a common mechanism (fig. S7 and Suppl. Note 4). However, alpha clocking power was higher in deeper layers as compared to superficial layers, possibly suggesting that the alpha clock originates in the deeper cortical layers.”

6. The authors cite a number of studies showing rhythmic attentional sampling fixed at one location. The implication is that this is the first study to show “attentional saccades” of the mind’s eye. It is not. Buschman reported virtually the same result in the FEF (Buschman and Miller, 2009 Neuron). This, by the way, predates the other work cited by the authors but it is not cited despite the seemingly similar results. Buschman also found rhythmic covert search (attentional saccades of the mind’s eye) that correlated with rhythms in the pre-frontal cortex. It was at a faster pace (18-24 Hz) but that is likely a trivial difference. The “clock” may run at different depending on task demands etc. I’m not saying that this precedent lessens the value of this work. The authors go beyond that prior work by showing effects of distractors, task demands etc. But the Buschman paper should not only be cited but also discussed given the similarity of the results. To not do so would make the current results seem more novel than they actually are. That is not good scholarship.

This study is well known to our group and we apologize for not citing it.

The following text has been added to the introduction: “Recent works propose that neural oscillations in the fronto-parietal network organize alternating attentional states or shifts in attention that in turn modulate perceptual sensitivity (Fibelekorn et al., 2018, 2019, Buschman & Miller, 2009).”

This work is also discussed as follows in the discussion section:

“Attentional periodicity in monkey FEF has been suggested before (Buschman & Miller, 2009), albeit with a faster frequency (18-25Hz). This frequency difference may reflect disparities in experimental design or task difficulty. Alternatively, it could reflect specific differences between FEF LFP and MUA processes (fig. S8a). The present study goes beyond this prior work by explicitly decoding the position of attention over time, and exploring the effect of distractors and task contingencies.”

It is also cited in the result section as follows:

“Previous studies indicate a coupling between LFPs theta frequency content and behavior (Fiebelkorn et al., 2018) as well as between LFP beta frequency content and spiking activity and behavior (Buschman and Miller, 2009). An important question is thus whether changes in super MUA are linked to the phase of oscillatory activity in the local field potentials. A significant alpha component is present in the super MUAs (fig. S8b) but not in the LFPs (fig. S8a). However, a significant phase-phase coupling can be identified between these two signals, in the alpha range (7-12Hz) as well as in the beta frequency range (18-30Hz, fig. S8d, suppl. Note 5). The functional significance of this coupling, its directionality and its causal relationship to attention and perception remains to be explored.”

Reviewer #3 (Remarks to the Author):

The present manuscript, titled “Prefrontal attentional saccades explore space rhythmically,” used multi-unit activity recorded in the prefrontal cortex (FEF) of macaques to track the locus of spatial attention during a cue-target delay. The results indicate, in line with other recent work, that spatial attention is a dynamic process, with sampling occurring rhythmically at a frequency in the range of 7-12 Hz. The authors use novel approaches to decode and track the attentional spotlight. Their results provide further support for a new, rhythmic characterization of spatial attention, and further evidence of the underlying neural basis. Below I provide some specific comments/questions:

We would like to thank reviewer #3 for his/her positive appreciation of our work and the very constructive comments he/she has provided us with. Below is a point by point response to his/her comments.

(1) Generally speaking, I do not think that the Summary fully conveys the novel aspects of the present manuscript. See the following comments, 2-4.

We thank reviewer #3 for his/her feedback on this crucial part of the manuscript. Our detailed answer to points 2 to 4 are developed below.

(2) In the Summary, “Recent studies suggest that attention samples space rhythmically through oscillatory interactions in the frontoparietal network. However, the precise mechanism through which prefrontal cortex, at the source of attention control signals, organizes this rhythmic exploration of space remains unknown.”

This precise mechanism does not seem to be defined in the summary. How specifically does the present manuscript go beyond other recent studies (Spyropoulos et al., PNAS, 2018; Fiebelkorn et al., Neuron, 2018; Helfrich et al., Neuron, 2018; Kienitz et al., Current Biology, 2018; Fiebelkorn et al., Nature Communications, 2019) at getting at the underlying mechanism?

We have now redrafted the Summary as follows. This version also includes our response to comments 3 and 4, and follows the 150 words count limit:

“Recent studies suggest that attention samples space rhythmically through oscillatory interactions in the frontoparietal network. How these attentional fluctuations coincide with spatial exploration/displacement and exploitation/selection by a dynamic attentional spotlight under top-down control is unclear. Here, we show a direct contribution of prefrontal attention selection mechanisms to a continuous space exploration. Specifically, we provide a direct high spatio-temporal resolution prefrontal population decoding of the covert attentional spotlight. We show that it continuously explores space at a 7-12Hz rhythm. Sensory encoding and

behavioral reports are increased at a specific optimal phase w/ to this rhythm. We propose that this prefrontal neuronal rhythm reflects an alpha-clocked sampling of the visual environment in the absence of eye movements. These attentional explorations are highly flexible, how they spatially unfold depending both on within-trial and across-task contingencies. These results are discussed in the context of exploration-exploitation strategies and prefrontal top-down attentional control.”

(3) In the Summary, I do not think the “the mind’s eye” is an appropriate metaphor for covert attention. The “mind’s eye” typically refers to mental imagery.

We have removed this reference to the mind’s eye. The summary now reads as follows:

“Specifically, we provide a direct high spatio-temporal resolution prefrontal population decoding of the covert attentional spotlight. We show that it continuously explores space at a 7-12Hz rhythm.

(4) In the Summary (and throughout the manuscript), “We propose that this rhythmic prefrontal attentional spotlight dynamics corresponds to a continuous overt exploration of space via alpha-clocked attentional saccades.” I disagree with this characterization or maybe just the wording. First, the use of overt to refer to something that’s happening covertly, when overt is typically used in this context to refer to exploratory movements. Similarly, attentional “saccades” is a misleading term, as “saccade” is defined as an eye movement. I think the authors need more clarity in their description, maybe “We propose that these rhythmic dynamics in prefrontal cortex reflect an alpha-clocked sampling of the visual environment that continues to occur in the absence of eye movements.”

We need to apologize here for a quite incredible typo: what we meant was obviously “We propose that this rhythmic prefrontal attentional spotlight dynamics corresponds to a continuous COVERT exploration of space via alpha-clocked attentional saccades.”

However, we do take Reviewer #3’s suggestion and now change this part of the summary as follows:

“We propose that this prefrontal neuronal rhythm reflects an alpha-clocked sampling of the visual environment in the absence of eye movements.”

(5) Similar to other recent papers, the authors link rhythmic changes in neural activity with changes in sensory processing and perceptual outcomes. The authors describe rhythmic changes in neural activity as supporting either exploitation or exploration. A recent TICS paper, by Fiebelkorn and Kastner (2019) describes rhythmic changes in neural activity as supporting either sampling or shifting. The authors should discuss how their own ideas are similar to or contrast with this previous, related work.

This is an excellent point. We are currently working at experimentally characterizing the functional link between our alpha-clocked rhythmic population attentional spotlight and theta LFP oscillations. This work is too preliminary to include here. However, we now have added the following paragraph to the discussion section on **Exploring versus exploiting space by attention**: “Fiebelkorn & Kastner (2019) propose that theta rhythms organize neural activity into alternating attentional states associated with either sampling (coinciding with periods of enhanced perceptual sensitivity) at a behaviorally relevant location or shifting to another location (coinciding with periods of decreased perceptual sensitivity). In this model, how much overt or covert attention is placed onto a given item of the visual scene depends on its behavioral relevance. This model provides a nice comprehensive interpretation of changes in the parieto-frontal neuronal synchronization and coupling properties during 80% validity cued target detection tasks. The task used here is a 100% validity task. There is thus no behavioral drive to explore uncued locations, as one would expect in 80% validity cued target detection tasks. And indeed, the decoded prefrontal attentional spotlight only rarely directly explores the uncued landmarks (fig. 9), though it does visit the different quadrants in a way that varies from one task configuration to another (fig. 10). This important task difference might actually account for the higher alpha content we observe in the decoded spatial attention spotlight traces as compared to the theta content (fig. S5). In this task configuration, we show that the attentional spotlight explores space at an alpha-clock pace, sometimes visiting the cued location (exploitation) and sometimes visiting uncued spatially irrelevant locations (exploration). Because exploitation is an unexpected low frequency event, we propose that exploration is the

default mode of the system, while exploitation, requires effort or a top-down drive to be implemented. Whether this exploitation is implemented by an independent theta clock remains to be tested. This would reconcile the seemingly contradictory views of the sampling/shifting hypothesis and an alpha exploration/exploitation hypothesis.”

(6) The authors used a behavioral task with 100% cue validity, which means there is no clear behavioral measure of attentional deployment. The authors do show that two different distractor types are associated with worse performance at the cued location. Here, I have two requests: first, the authors use two different types of distractors but do not differentiate between them when reporting how they influence behavioral performance (Figure 1B). Please report performance in the presence of distractors separately for the distractor types. Second, could the authors please provide some basic neurophysiological evidence of attentional deployment at the cued location? The authors could provide MUA during the cue-target interval when response fields overlapped the cued location relative to when response fields overlapped a non-cued location.

The behavioral response to the two types of distractors is now presented in fig. S1. This figure is now cited in the result section as follows: “Distractors were presented during the cue-to-target interval and target luminosity was adjusted so as to make the task difficult to perform without orienting attention (fig. 1b, fig. S1 independently reports the behavioral performance for both distractors at the landmarks and distractors in the workspace, fig. S2 and supplementary Note 1 report the MUA spatial attention selectivity across recording sessions).”

Figure S1 and its legend are reproduced below.

Figure S1: **Behavioral performance of monkeys M1 and M2 for both distractors at the landmark (LM) and distractors in the workspace (WS)**, compared with target detection in the absence (No) of a distractor (median % correct +/- median absolute deviation).

The MUA response characterization to a cue in the receptive field (preferred response position) versus outside (non-preferred response position) the receptive field of the MUA in now presented in fig. S2. This figure is now cited in the result section as follows: “In order to access FEF attentional content in time, monkeys performed a manual response cued target-detection task (fig. 1a) while we recorded the MUA bilaterally from their FEF neuronal ensembles, using two 24-contacts recording probes (fig. 1c). Distractors were presented during the cue-to-target interval and target luminosity was adjusted so as to make the task difficult to perform without orienting attention (fig. 1b, fig. S1 independently reports the behavioral performance for both

distractors at the landmarks and distractors in the workspace, fig. S2 and supplementary Note 1 report the MUA spatial attention selectivity across recording sessions).”

Figure S2 and its legend are reproduced below.

Figure S2: **MUA spatial attention selectivity.** (a) Single MUA mean (+/- s.e.), when cue is orienting attention towards the preferred (black) or the anti-preferred (gray) spatial location, during the cue to target interval. X-axis represents time around the cue to target interval. (b) MUA spatial attention selectivity for a representative recording session. X-axis represents time around the cue to target interval. Y-axis represents individual channels, separated in left and right hemisphere channels. Each line represents, for each individual channel, the difference between the normalized neuronal response to a cue orienting attention towards the preferred spatial location and the normalized neuronal response to a cue orienting attention towards the anti-preferred

spatial location. White ticks represent the onset of statistically significant differences between these two signals (Wilcoxon, $p < 0.05$). (c) Distribution of a spatial attention index (Preferred-AntiPreferred)/(Preferred+AntiPreferred), computed over [-200 0] ms before target onset) across all MUA of all sessions. Red histogram corresponds to channels in which the neuronal activity during this time interval was significantly different between the preferred and the anti-preferred spatial attention responses (Wilcoxon, $p < 0.05$, gray, no significant difference).

The following information is also now added to the supplementary information:

“Note 1: Description of the neuronal population response properties

The recorded receptive fields are quite large, as typically described in the FEF. Sixty-one percent of the MUA channels had a significant ***target related*** response on correct trials. Of these, 23.5% of the recorded RFs encompass one visual quadrant, 24.2% encompass two ipsilateral visual quadrants, 4.6% encompass two opposing visual quadrants, 21.6% encompass three visual quadrants and 26.1% encompass 4 visual quadrants. Seventy-three percent of the MUA channels had a significant ***attention related*** response on correct trials. Of these, 14.1% of the recorded RFs encompass one visual quadrant, 14.4% encompass two ipsilateral visual quadrants, 3.6% encompass two opposing visual quadrants, 20.2% encompass three visual quadrants and 47.7% encompass 4 visual quadrants.

This diverse receptive field structure of the data was critical for the success of the linear decoding approach that we are using here. Noteworthy is the fact that, in addition to significantly modulated neurons, non-significantly modulated neurons also contributed to the decoder (Farbod Kia et al. 2011).

Fig. S2 further reports the MUA spatial attention selectivity on an exemplar MUA signal, an exemplar session and across recording sessions.”

(7) To what extent does the averaging window for decoding influence the observed frequency of the attentional sampling? The 50-ms averaging window is equivalent to a 20-Hz sampling rate, which means the Nyquist frequency is 10 Hz. The authors should not report findings at any frequencies above 10 Hz, yet the figures (e.g., Figure 2B) show results from 0-55 Hz. The averaging window is acting as a low-pass filter at higher frequencies.

This is a very interesting and relevant question. This is now addressed in the supplementary information as follows. A methodological manuscript formalizing this point is under preparation.

“Note 2: Averaging filter impact on decoded signal frequency content

Independent time series of our data (for each channel and each trial) are averaged over 50ms time windows (moving averaging filter of frequency $f_a=20\text{Hz}$) at every 1ms time step (sampling frequency $f_s=1\text{kHz}$). This results in a continuous time series (fig. 8). These rhythmic time series can be modelled by artificial attention signals (fig. S4a), the frequency content of which is strongly attenuated at f_s , $2f_s$, $3f_s$ etc. (fig. S4d, light gray).

In the decoding procedure described in the present manuscript, the rhythmicity of spatial attention is not only captured by individual channel changes in spiking probability, but also in the population code, due to specific co-activation patterns across the entire population that are captured by the decoding procedure. Fig. S4 models this point. Fig. S4a (left panel), represents an artificial attentional signal sampling space at 24Hz, in time (x-axis), over multiple trials (y-axis). From one trial to the next, the encoded spatial position of attention is different. The 24Hz frequency is higher than the moving averaging filter central frequency (20Hz). For each trial ($n=150$), 50 independent time series (representing the spiking activity of 50 independent MUA channels), were generated from the corresponding artificial attentional signal as follows. Each channel was associated with a specific spiking probability that varied as a function of the artificial attention signal (position of attention). For each trial, spatial attention was then decoded from this population (fig. S4a, right panel). Fig. S4b (blue) represents such decoded spatial attention information on a representative trial. It captures both frequency specificity and power (fig. S4c) of the input rhythmic spatial attention signal on the same trial (fig. S4b, red).

Fig. S4d generalizes these observations to input rhythmic signals ranging from 3 to 70Hz. The black trace corresponds to the actual input power frequency relationship in our artificial data. The light gray curve represents the recovered frequency content from the moving average filtered input independent time series ($f_a=20\text{Hz}$). As predicted by signal processing theory, frequencies at f_a and multiples of f_a cannot be recovered. The intermediate gray trace represents the frequency content recovered from the populational decoding procedure. While this procedure leads to an attenuation of the power of the recovered frequencies that is comparable to that observed on the moving averaging filtered input data, frequencies can be recovered along the entire test frequency range, including at frequencies higher than f_a , and at all f_a harmonics. This is due to the fact that, the rhythmicity of spatial attention is not only captured by individual channel changes in spiking probability, but also in the population code. While the moving averaging filter cancels f_a and f_a harmonic frequencies, the 1ms sampling procedure generates specific co-activation patterns across the entire population. These co-activation patterns are captured by the decoding procedure. In our simulations full recovery of f_a frequency content is achieved for a number of channels of ten or more (data not shown).

Importantly, all frequency contents reported in the present manuscript are identified against a 95% C.I. generated from random permutation of the data prior to the decoding procedure. As a result, this chance level captures the frequency attenuation described in fig. S4d."

Figure S4 is duplicated below:

Figure S4: **Influence of moving average filtering procedure on decoded signal frequency content.** (a) Artificial spatial attention signal exploring left/right space rhythmically at 24Hz, in time (x-axis), across multiple trials (y-axis), left panel. Fifty independent times series (representing the spiking activity of 50 independent MUA channels), are generated from the corresponding artificial attentional signal, each associated with a specific

spiking probability that varied as a function of the artificial attention signal (position of attention). On each trial, the times series of these 50 channels are used for decoding the spatial position of attention in time (right panel). (b) Decoded spatial attention information (blue) and input rhythmic spatial attention signal (red), on the same representative trial. (c) Frequency analysis of decoded spatial attention time series (blue) and input rhythmic spatial attention time series (red). (d) Generalization of decoder attenuation profile on frequencies ranging from 3 to 70Hz (intermediate gray) compared to single channel attenuation profile after moving average filtering ($f_a=20\text{Hz}$). Black: test frequency power relationship.

This is cited in the main result section as follows: “These fluctuations are reliably associated with a distinct peak in the power spectrum relative to chance, in the 7 - 12 Hz range (fig. S4 and suppl. Note 2 for a discussion impact of averaging filter on decoded signal frequency content).”

(8) “The decoder was trained... at a specific time during the cue-target delay,” please provide clarification here. What specific time? If the attentional spotlight is dynamic, then the specific time point relative to the cue should influence how the decoder is classifying the locus of attention (i.e., did that time point fall at a peak or a trough in the sampling rhythm). The primary point of the paper is that the attentional spotlight is highly dynamic, shifting around the visual scene, but shouldn't that make it difficult then to train the decoder in the first place?

This is an excellent question! This is now clarified in a dedicated paragraph in the supplementary material, entitled “Applying classification procedures to the decoding of a dynamic attention spotlight”, that reads as follows:

“Note 6: Classification procedures applied to a dynamic process

Classical classification approaches applied to the decoding of cortical activity are designed to associate specific populational signatures with specific experimental task components or classes. The essence of these approaches is to identify, based on a set of exemplar data (training set), for each class of interest, common response patterns (noise) irrespective of potential inter-trial variability (noise), and to evaluate the efficacy of the identified common response patterns on a novel data set (testing set). In our approach, we push this approach one step further, considering that on each trial, part of the noise (i.e. part of the distance between the observed response pattern and the actual corresponding class response pattern) is actually signal and characterizes attentional dynamics. In other words, we apply machine learning to neuronal signals under the assumption of stability of the attentional spotlight and then we interpret decoding error to the expected class as a signature of attentional dynamics. This is possible because these types of classifiers are driven by both the mean and the variability around the mean of the test class data. The relevance of this approach is confirmed by the fact that, while this might seem counter intuitive, this approach actually proves extremely efficient in capturing both specific neuronal and behavioral processes and accounting for an important part of observed variability (e.g. neuronal response to target and distractor, variations in hit rate, variations in false alarm rates, fig. 4 & 7).

Post-cue cross-temporal decoding maps capture the rhythmic nature of attention. This is due to the fact that the cue resets the attentional rhythm. If attention exploration was rhythmic, yet completely random over space, the resultant cross-temporal decoding map wouldn't be expected to show any clear rhythmicity. However, we show that attention expresses predictable exploration patterns between the cued and uncued quadrants (see fig. 10). These transitions of the attentional spotlight between quadrants fully account for the observed rhythmic oscillations in the post-cue cross-temporal decoding maps.”

This paragraph is mentioned in the Methods section as follows:

“The decoder was trained on a random set of 70% of the correct trials at all times in the cue to target interval, then tested on the 30% remaining at all time after cue presentation (see suppl. Note 6 for a discussion of how classical decoding techniques apply to the decoding of a dynamic attentional spotlight as described here).”

(9) The sampling frequency observed in the present results is a bit higher than the sampling frequency described in some of the previous work. Might this somewhat higher sampling frequency (at ~9 Hz) reflect the

specific experimental design, i.e., 100% cue validity? Such conditions might influence how/whether the attentional spotlight is split across multiple locations (see Landau and Fries, *Current Biology*, 2012 and Re et al., *Current Biology*, 2019).

We thank reviewer 3 for this question, which is now addressed in the end of the discussion section “**The prefrontal attentional spotlight explores space rhythmically**” as follows:

“Several recent behavioral studies strongly suggest that attention fluctuates at around 8 Hz. This sampling can be distributed across multiple spatial locations (Landau and Fries, 2012, Holcombe et al. 2013) or multiple objects at a given location (Re et al., 2019). Here, we argue that the decoded attentional spotlight provides a direct access to the intrinsic attentional rhythm, i.e. 8Hz, though how this reflects onto behavior will fully depend on task design and on how the spotlight successively samples relevant task locations. In our task, cues have a 100% validity. Hence, the attentional spotlight is on average into the cued quadrant, where attentional sampling, as assessed behaviorally, takes place at 8Hz. We predict that if the cue was not fully valid, behavioral sampling frequency might be lower than 8Hz, directly co-varying with cue validity.”

(10) In the results, “Oscillations in the attention-related population activity can either reflect a global rhythmic entrainment of the entire FEF population or changes in the FEF population code at a specific frequency.” Would the authors unpack this a bit, I didn’t understand the difference between these possibilities until I read further. Also please define “super MUA” in the main text of the paper (it is already defined in the methods). Apologies if I missed the definition in the main text.

We now rephrase the beginning of this paragraph to clarify what we mean by this sentence. Following suggestion of reviewers #1 & #3, we also redefine “super MUA” in the result section. We now also provide a more extensive justification of using the super MUA measure in response to the comments of reviewer #2. This justification reads as follows:

“Oscillations in the attention-related population activity can either reflect a global rhythmic entrainment of the entire FEF population (i.e. all spiking rates of all neurons throughout the FEF, changing their firing rates synchronously, being enhanced or suppressed coherently) or changes in the FEF population code at a specific frequency (i.e. the spiking rates of only some FEF neurons changing their firing rates, being enhanced or suppressed, at any peak or trough of the identified oscillations, each specific neuronal combination corresponding to a specific spatial attentional code). Fig. 3A represents, for one exemplar recording probe, on an exemplar trial, and for each recording channel, the time epochs at which spiking-rate exceeds the 65% of the maximum spiking regime of the individual channel. On every single channel, these high spiking probability epochs do not appear to follow a systematic rhythm, thus contradicting a global rhythmic entrainment hypothesis. Rather, this high spiking probability organizes in discrete epochs, distributed over all recording channels. The hypothesis that changes in the FEF population code (and thus high spiking probability epochs) take place at a specific frequency implies that average MUA over all channels on a given trial will show marked rhythmic variations in firing rate in time. Fig. 3b confirms this hypothesis. For this individual trial, a *super MUA* signal was computed by averaging the spiking activity of the 48 recording channels on this specific trial. Peaks of the alpha oscillations are clearly identified on the super MUA of the same individual trial³⁸ (fig. 3b) and plotted against the spiking probability changes represented in fig. 3a. The high spiking probability epochs of individual channels coincide with peak alpha oscillatory phases in the super MUA. This is captured by a spectral analysis of changes in spiking probability in a frequency range running from 5 to 15 Hz (see material & methods). Most channels of fig. 3a display a modulation of spiking probability in an 8 - 12 Hz frequency range (fig. 3c, color code matching fig. 3a). This holds true for all sessions (fig. 3d, mean+/-s.e.). However, this alpha rhythmic modulation of spiking probability does not reflect a global entrainment of the entire population. This can be seen in fig. 3a in which individual channels do not exhibit on any given trial, high spiking rate synchronously at each identified alpha cycle. This is also captured in figure 3c, as the degree of alpha locking of spiking activity varies from one channel to the next. Rather, the channels with highest change in normalized spiking activity change from one super MUA alpha peak to the next, thus reflecting a change in the FEF population code. These variations correspond to changes in the spatial allocation of the attentional spotlight that will be described hereafter.”

(11) It is very difficult to see the alpha-filtered “super MUA” in Figure 3B.

This has now been fixed, as illustrated below:

Figure 3: **Alpha rhythm paces FEF population code.** (a) Individual channel spiking probability at a threshold of 65% (1 trial, 48 channels) in time. (Cue is presented at 700ms. Grey vertical lines: peak of alpha cycles of the super MUA in (b). Individual channels ordered and color coded in a gradient of blue, as a function alpha locking amplitude in (c). (b) Raw (black trace) and alpha filtered (blue trace) single trial population super MUA calculated over the 48 MUA channels. Grey vertical lines: peak of alpha cycles of the super MUA. (c) Changes in individual channel spiking probability, across all trials, as a function of putative locking to frequencies from 5 to 15 Hz. Spiking probability is specifically affected in the alpha frequency. Channels color coded in a gradient of blue, as a function of alpha locking amplitude. (d) Mean \pm s.e. phase frequency modulation of spiking activity across all sessions and all channels.

(12) The authors demonstrate evidence of oscillatory patterns in the behavioral data, with peaks in both the theta and alpha ranges. Is there any evidence of a relationship between theta and alpha in the neural data (e.g., Fiebelkorn et al., Nature Communications, 2019)?

This following text is now added to the result section as is, following the discussion of fig. 2b.

“Fig. S5 represents the average normalized power spectrum of spatial attention classification (same time window as in fig. 2b) for all sessions. Two significant local maxima can be identified against the 95% confidence interval, one in the theta range and one in the alpha range. The alpha peak is higher than the theta peak and coincides with the attentional rhythms discussed in the present work. Importantly, while the alpha attention information oscillation can be identified in all sessions (100%), the theta oscillation can only be identified in 68% of the sessions (13/19 sessions, significance assessed against the 95% confidence interval). In the following, we focus on the higher and most significant frequency peak, namely the alpha power.”

Reference to this figure S5 is also added to the discussion of hit rate rhythmic variations as follows:

“... two significant oscillatory peaks are observed onto behavior, one in the theta (3 to 5 Hz) frequency band, and one in the alpha (9 to 14 Hz) frequency band (fig. 5B), thus reproducing previous behavioral observations^{20–24,37}. These two peaks coincide with those identified in the prefrontal attention-related information (fig. S5), as well as with those identified in the FEF LFPs (Fiebelkorn et al., 2018). This alpha peak expresses in a frequency range that is lower than the FEF-theta locked alpha peak identified in the pulvinar (Fiebelkorn et al., 2019).”

Figure S5 is replicated below:

Figure S5: **Averaged peak oscillations of prefrontal attention-related information across all session.** Normalized power identified in the cross-temporal classification reference interval (fig. 2b), (dark grey: significant frequencies relative to 95% C.I.) for all sessions (4-cued task version).

(13) Did the authors measure whether changes in MUA are linked to the phase of oscillatory activity in the local field potentials?

This is an excellent question that would allow to relate our current observations to previous literature based on correlational analyses between behavior and LFPs. This analysis is ongoing, and is less straightforward than what one might expect due to the fact that it aims at relating a distributed population measure (multi-channel MUA decoding) with more local mesoscopic measures (LFP).

We do find a PPC spike field coherence peak at alpha frequency range (~8Hz). However, at this stage, this data is too preliminary to report as such.

We now however report a more global measure of phase relationship between the super MUAs (reflecting the general alpha-clocking of the neuronal population) and the individual LFPs (reflecting both long range inter-areal and local processes) on each session. This is presented in a new supplementary information section as follows:

“Note 5: Phase relationship between super MUA and LFPs

The super MUAs reflect the general alpha-clocking of the entire bilateral FEF neuronal population. This activity is phase locked between the two hemispheres (fig. S6 and related supplementary information). In contrast, individual LFPs reflect both long range inter-areal and local processes on each session. Fig. S8 represents the frequency content (mean +/- s.e.) of both the task-related LFPs (fig. S8ab) and the super MUAs (fig. S8c) across all sessions. LFPs show an enhanced frequency content in the lower theta range (3-5Hz) as well as in the beta frequency range (18-30Hz). These observations are consistent with previous reports (Fiebelkorn et al., 2018, Buschman and Miller, 2009). In contrast, while the theta peak of the super MUAs is weak, these signals show a marked frequency peak in the alpha range (7-12Hz, coinciding with the frequency range described in the present work, Fig. 2 & 3), as well as a consistent peak in the beta frequency range (18-30Hz). Alpha oscillatory mechanisms thus appear to be specific to the super MUAs. Fig. S8d represents phase-phase coherence between the LFP and super MUA signals. Coherence is enhanced in the three frequency bands identified in fig. S8ab, namely the lower theta range (3-5Hz), the alpha range (7-12Hz), as well as in the beta frequency range (18-30Hz). Overall, this thus suggests a strong phase coupling between the FEF LFPs and super MUAs. The

functional significance of this coupling, its directionality and its causal relationship to attention and perception remains to be explored.”

This section is now referenced in the main result section as follows:

“However, this alpha rhythmic modulation of spiking probability does not reflect a global entrainment of the entire population. [...] Rather, the channels with highest change in normalized spiking activity change from one super MUA alpha peak to the next, thus reflecting a change in the FEF population code. These variations correspond to changes in the spatial allocation of the attentional spotlight that will be described hereafter. Super MUAs independently computed across left and right electrodes on individual trials oscillated at a common rhythm (fig. S6a) as well as at a common phase (fig. S6b, suppl. Note 3). This contrasts with the fact that decoded population attention information also oscillated at a common rhythm (fig. S6c) when computed independently on the left and right probes, but in anti-phase one with respect to the other (fig. S6d). This confirms that these variations in MUA spiking probability correspond to changes in the spatial allocation of the attentional spotlight and suggest an active inter-hemispheric coupling mechanism.

[...]

Previous studies indicate a coupling between LFPs theta frequency content and behavior (Fiebelkorn et al., 2018) as well as between LFP beta frequency content and spiking activity and behavior (Buschman and Miller, 2009). An important question is thus whether changes in super MUAs are linked to the phase of oscillatory activity in the local field potentials. A significant alpha component is present in the super MUAs (fig. S8b) but not in the LFPs (fig. S8a). However, a significant phase-phase coupling can be identified between these two signals, the alpha range (7-12Hz), as well as in the beta frequency range (18-30Hz, fig. S8d, suppl. Note 5). The functional significance of this coupling, its directionality and its causal relationship to attention and perception remains to be explored.”

Figure S8 is replicated below:

Figure S8: **Alpha phase coherence between super MUAs and LFPs.** (a) Averaged LFP power spectrum (mean +/- s.e across all channels and sessions, dark grey: significance w/ 95%CI). (b) Close-up on low-frequency inset in (a), after 1/f correction. (c) Averaged super MUA power spectrum (mean +/- s.e, across all channels and sessions, dark grey: significance w/ 95%CI). (d) Averaged coherence between LFP and super MUA power spectra (mean +/- s.e, across all channels and sessions, dark grey: significance w/ 95%CI).

sessions, dark grey: significance w/ 95%CI). (d) Averaged phase coherency between super MUAs and LFPs, calculated using pairwise phase consistency (PPC) (mean \pm s.e, across all channels and sessions, dark grey: significance w/ 95%CI).

Reviewers' Comments:

Reviewer #1:

Remarks to the Author:

The reviewers have extensively answered my comments and I have no further comments. In my view, this is an excellent manuscript that should be granted publication.

Reviewer #2:

Remarks to the Author:

The reviewers have address my concerns to my satisfaction.

Reviewer #3:

Remarks to the Author:

I am satisfied with the authors responses to all of my comments. I fully support the publication of this manuscript.